# Genetic investigation of fibromuscular dysplasia identifies risk loci and shared genetics with common cardiovascular diseases

Adrien Georges [1,32], Min-Lee Yang[2,3,32], Takiy-Eddine Berrandou [1,32], Mark K. Bakker [4], Ozan Dikilitas [5], Soto Romuald Kiando[1], Lijiang Ma[6], Benjamin A. Satterfield[5], Sebanti Sengupta[2], Mengyao Yu[1], Jean-François Deleuze [7], Delia Dupré[1], Kristina L. Hunker[2,3], Sergiy Kyryachenko[1], Lu Liu[1], Ines Sayoud-Sadeg[1], Laurence Amar[1,8], Chad M. Brummett[9], Dawn M. Coleman[10], Valentina d'Escamard[11], Peter de Leeuw [12,13], Natalia Fendrikova-Mahlay[14], Daniella Kadian-Dodov[15], Jun Z. Li [3], Aurélien Lorthioir[1,8], Marco Pappaccogli[16,17], Aleksander Prejbisz[18], Witold Smigielski[19], James C. Stanley[10], Matthew Zawistowski [20], Xiang Zhou [20], Sebastian Zöllner[20], FEIRI investigators*, International Stroke Genetics Consortium (ISGC) Intracranial Aneurysm Working Group*, MEGASTROKE*, Philippe Amouyel [21], Marc L. De Buyzere[22], Stéphanie Debette [23], Piotr Dobrowolski[18], Wojciech Drygas[24], Heather L. Gornik [14], Jeffrey W. Olin [15], Jerzy Piwonski[24], Ernst R. Rietzschel[22], Ynte M. Ruigrok [5], Miikka Vikkula [25], Ewa Warchol Celinska [18], Andrzej Januszewicz[18], Iftikhar J. Kullo [5,26], Michel Azizi[8,27], ARCADIA Investigators*, Xavier Jeunemaitre [1,28], Alexandre Persu[16,29], Jason C. Kovacic[15,30,31], Santhi K. Ganesh [2,3,33 ✉] & Nabila Bouatia-Naji [1,33 ✉]

Fibromuscular dysplasia (FMD) is an arteriopathy associated with hypertension, stroke and myocardial infarction, affecting mostly women. We report results from the first genome-wide association meta-analysis of six studies including 1556 FMD cases and 7100 controls. We find an estimate of SNP-based heritability compatible with FMD having a polygenic basis, and report four robustly associated loci (*PHACTR1, LRP1, ATP2B1,* and *LIMA1*). Transcriptome-wide association analysis in arteries identifies one additional locus (*SLC24A3*). We characterize open chromatin in arterial primary cells and find that FMD associated variants are located in arterial-specific regulatory elements. Target genes are broadly involved in mechanisms related to actin cytoskeleton and intracellular calcium homeostasis, central to vascular contraction. We find significant genetic overlap between FMD and more common cardio-vascular diseases and traits including blood pressure, migraine, intracranial aneurysm, and coronary artery disease.

---

A full list of author affiliations appears at the end of the paper.

Cardiovascular disease (CVD) is the primary cause of mortality in the world. CVD causes ~18 million deaths each year, of which 85% are due to stroke and myocardial infarction (MI)[1]. Renal artery stenosis is a cause of hypertension, a preventable risk factor for stroke and MI. Renovascular hypertension results from numerous factors, which include atherosclerosis or fibromuscular dysplasia (FMD) in ~10% of cases[2]. While atherosclerosis has been widely studied and its genetic architecture has been well defined, little is known about the pathogenesis or genetics of FMD. To date, only *PHACTR1*, a pleiotropic locus involved in the genetic risk of several cardiovascular and neurovascular diseases, has been reported to associate with FMD[3].

FMD occurs predominantly in early middle-aged women (mean age at diagnosis 46–53 years)[4], thus representing a subset of the population where cardiovascular and neurovascular disease present differently depending on sex[5,6]. FMD is an idiopathic, segmental, non-atherosclerotic disease of the arterial walls, leading to stenosis of small and medium-sized arteries, often associated with dissection, aneurysm, and in some cases arterial tortuosity[4,7–9]. The most frequent form of FMD is multifocal, which is characterized by multiple stenoses occurring in the same artery (so called "string of beads" due to classic angiographic appearance), and accounts for 70–80% of cases[4]. Further mention of FMD in the current report will refer to the multifocal type. Diagnosis is often made incidentally on imaging, as part of an investigation to elucidate early onset and/or resistant hypertension, following a stroke event or spontaneous coronary artery dissection, a form of acute MI associated with female sex as well[10]. Based on angiographic examination of renal arteries in kidney donor cohorts the estimated prevalence of FMD is ~3%[11].

In the current study, we report findings from a meta-analysis of six genome-wide association studies (GWAS) to investigate the genetic basis of FMD. We report four loci (*PHACTR1*, *LRP1*, *ATP2B1*, and *LIMA1)* that are associated with FMD, in addition to *SLC24A3* identified through transcriptome-wide association studies of arterial transcriptomes from the GTEx database. Through the integration of open chromatin maps generated by ATAC-Seq in vascular cells, combined with publicly available data for arteries, we prioritize variants in associated loci. Using colocalization with expression quantitative trait loci (eQTLs) and transcriptome-wide association studies in arteries, we narrow down potential target genes. We find that risk genes for FMD are consistently expressed in smooth muscle cells, fibroblasts, and arterial tissues. These genes are involved in regulatory mechanisms related to actin cytoskeleton and intracellular calcium homoeostasis, a mechanism central to vascular contraction. Finally, using linkage disequilibrium score regression, we report an estimate of single nucleotide polymorphism (SNP)-based heritability compatible with FMD having a polygenic basis, and an important genetic overlap with blood pressure, migraine, intracranial aneurysm, and coronary artery disease (CAD).

## Results

**Meta-analysis of six genome-wide association studies reveals four risk loci for FMD**. We tested ~5.5 million common genetic variants (MAF > 0.01), including those on the X chromosome, in six case control studies from Europe and the United States totalling 1556 cases and 7100 controls, all of European ancestry (Supplementary Table S1). Individual studies were adjusted for sex, the first five principal components, and genomic control. Using LD score regression[12], we found evidence for a polygenic feature for FMD with an estimated SNP-based heritability ($h^2$) on a liability scale of 0.43 (standard error: 0.14, estimated prevalence of 3% in the general population).

Four loci were associated with FMD at genome-wide significance (Fig. 1a, Supplementary Fig. S1, Supplementary Table S2). We replicated the previously identified *PHACTR1* locus on chromosome 6 (SNP rs9349379, OR = 1.44, $P = 5 \times 10^{-15}$) and reported three additional loci: *LRP1* (rs11172113, OR = 1.34, $P = 2 \times 10^{-10}$), *LIMA1* (rs7301566, OR = 1.29, $P = 4 \times 10^{-9}$) and *ATP2B1* (rs2681492, OR = 1.43, $P = 2 \times 10^{-8}$, Table 1, Supplementary Table S2). All lead SNPs showed consistent directions of effects and lacked evidence for heterogeneity (Table 1). Despite mapping to the same chromosome, *LIMA1, LRP1* and *ATP2B1* loci are fully independent and map to positions 50.5, 57.5 and 90.1 megabases on chromosome 12, respectively. A GWAS in women only (1421 cases, 5724 controls) identified the same four loci, with comparable effect sizes and levels of significance (Supplementary Table S2). Given the small number of FMD cases in men (N = 135), a GWAS in men only was not conducted.

To address the limitation of SNP-based GWAS, which captures the effects of single variants and may miss the combined mild effects of several SNPs in a gene, we performed gene-based association analyses using MAGMA[13], as implemented in FUMA[14]. We identified five genes from two of the previously identified loci to be associated with FMD ($P_{Bonf.Adj} < 0.05$) (Supplementary Table S3). These included four genes at the *LIMA1* locus (*ATF1*, $P = 4 \times 10^{-7}$, *GPD1*, $P = 5 \times 10^{-7}$, *CERS5*, $P = 1 \times 10^{-6}$, *LIMA1*, $P = 2 \times 10^{-6}$) and *ATP2B1* ($P = 5 \times 10^{-7}$). All genes were located in the vicinity of the 4 loci we identified using SNP-based GWAS, and no additional genes were identified using the gene-based approach.

**FMD-associated variants are likely to regulate gene expressions of nearby genes in arterial tissues**. We queried the GTEx database for eQTL association (v8 release)[15] and found the lead SNPs in FMD loci to be top eQTLs of at least one gene in one or more of the three arterial tissues available (Fig. 2a, Supplementary Table S4). Top eQTL associations linked lead variants to the closest gene at *PHACTR1*, *LRP1* and *ATP2B1* loci (Fig. 2b, d, h), with a high probability (100%, 100% and 98.3%, respectively) of colocalization between FMD and tibial artery eQTL association signals (Fig. 2c, e, i). FMD risk alleles were associated with higher expression of *PHACTR1* and *LRP1* and lower expression of *ATP2B1* in tibial artery tissue (Fig. 2b, d, f). We note that the expression of *ATP2B1* antisense transcript (*ATP2B1-AS*) was also regulated by lead SNPs at the *ATP2B1* locus in tibial artery samples, also with a high probability (96.6%) of colocalization (Supplementary Fig. S2). At the *LIMA1* locus, several genes were associated with the lead variant in artery tissues (Fig. 2a, f), but none of the eQTL signals colocalized definitively with the FMD association signal (Fig. 2g, Supplementary Fig. S2).

Interestingly, a query of transcriptome sequencing data obtained in primary dermal fibroblasts cell lines from 83 FMD patients[16] found rs9349379 to correlate with *PHACTR1* expression (P = 0.01), confirming the association observed in GTEx artery tissue (Supplementary Fig. S3).

**Transcriptome-wide association study in arteries**. We conducted a transcriptome-wide association study (TWAS) using the FUSION software[17], and gene expression models calculated from artery eQTL analyses from GTEx (v7 release) combining expression data from tibial, aorta and coronary arteries (Fig. 1c, d). We found significant association between genetically predicted expression of *PHACTR1* ($P = 1.1 \times 10^{-11}$, tibial and aorta), *LRP1* ($P = 2.7 \times 10^{-10}$, tibial) and FMD (Table 2, Fig. 1c, d, Supplementary Table S5). Genetically predicted expressions of *ATP2B1* and *ATF1* were suggestively (FDR < 0.05) associated with FMD

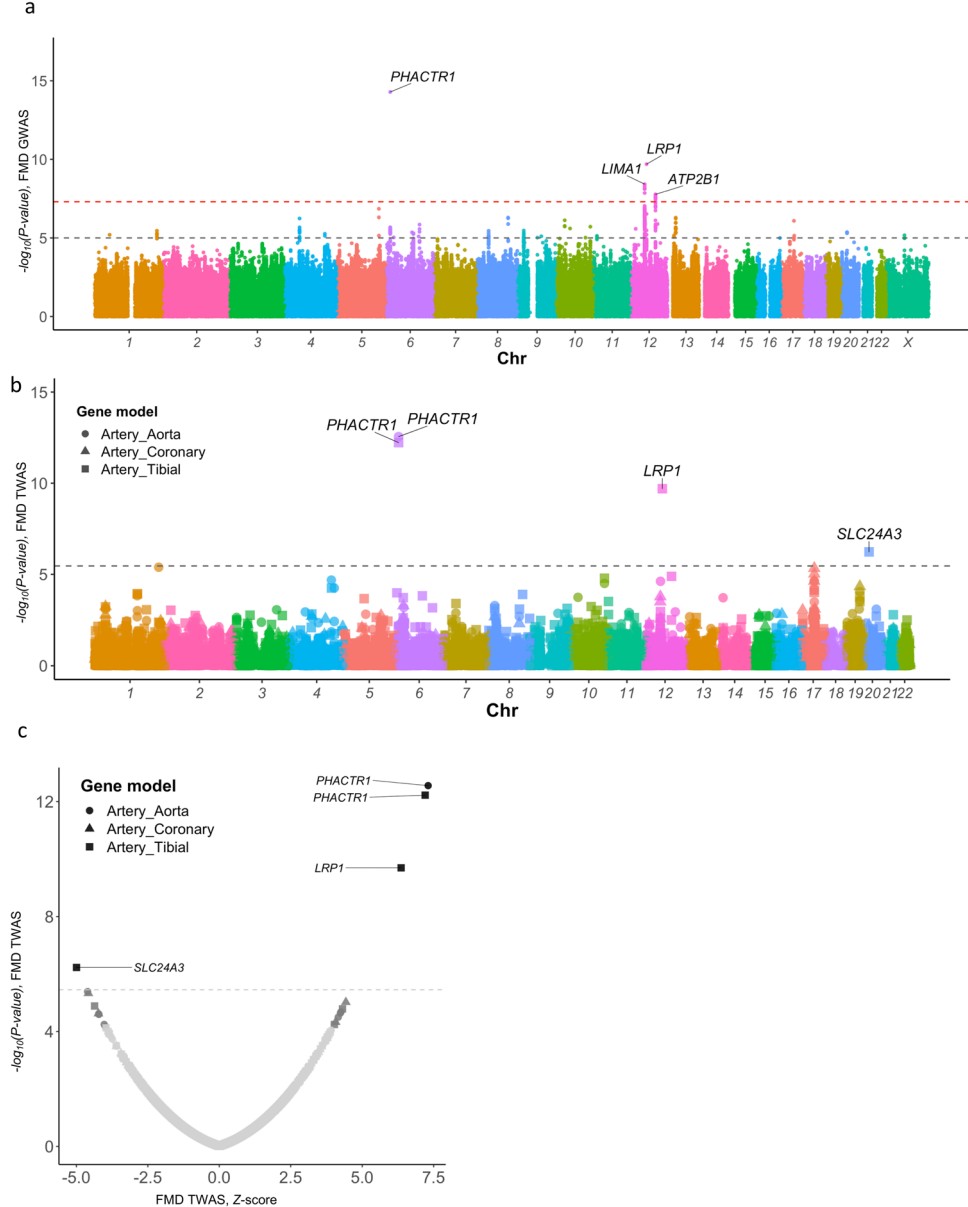

**Fig. 1 SNP-based and transcriptome-wide association analyses. a** Manhattan plot representation of SNP-based association analysis in FMD. $-\log_{10}$ of association $P$ value (from a two-sided Wald test) is represented on the $y$-axis, genomic coordinates on the $x$-axis. Name of lead SNPs with $P$ value $\leq 5 \times 10^{-8}$ are indicated. **b** Manhattan plot representation of Transcriptome-wide association analysis (TWAS) in FMD with tibial artery (square dots), aorta (round dots) and coronary artery (triangle dots) gene expression models. $-\log_{10}$ of association $P$ value is represented on the $y$-axis, genomic coordinates on the $x$-axis. Name of genes with Bonferroni corrected $P$ value $\leq 0.05$ are indicated. **c** Volcano plot representation of FMD TWAS. TWAS $Z$-score is represented on the $x$-axis, $-\log_{10}$ of TWAS $P$ value on the $y$-axis. The dashed line represents the threshold for significance adjusted for multiple testing. Name of genes with Bonferroni adjusted $P$ value < 0.05 are indicated.

(Supplementary Table S5). TWAS in tibial artery revealed *SLC24A3* to associate with FMD ($P = 6 \times 10^{-7}$, Fig. 1c, d). Of note, *SLC24A3* showed nominal significance in the gene-based analysis ($P = 2 \times 10^{-3}$), and it overlaps two independent and suggestive signals (lead SNP: rs2424245, $P = 4.0 \times 10^{-6}$, secondary SNP: rs6046121, $P = 4 \times 10^{-4}$, correlation $r^2 < 0.01$, Supplementary Fig. S4). FMD risk alleles of both variants associated with lower *SLC24A3* expression in tibial artery and aorta (Supplementary Fig. S4).

**FMD-associated loci are located in regulatory elements in arteries and VSMCs**. To gain insights into the potential regulatory function of the topmost associated SNPs with FMD, we

generated open chromatin profiles using ATAC-Seq in human carotid artery-derived primary cells (two VSMCs and two ECs primary cell lines), human coronary artery-derived cells (one VSMC and one EC primary cell line), human dermal (two cell lines) and cardiac fibroblasts (one cell line). We also reanalysed published data obtained using ATAC-Seq of healthy coronary arteries[18]. Using a common pipeline analysis, we obtained 177,015–196,272 peaks from cultured human VSMCs, 120,577–137,779 from ECs and fibroblasts, and 54,622–70,855 peaks from human coronary arteries (Supplementary Fig. S5a). Global correlation and principal component analyses showed that open chromatin regions of fibroblasts and VSMCs cluster together, whereas ECs and artery samples form separate clusters (Supplementary Fig. S5b, c). Using the GREGOR algorithm[19], we

**Table 1 FMD-associated variants in SNP association analyses.**

| rsID (Chr:pos) | Locus | EA | EAF | ARCADIA (431/1487) | | ARCADIA-POL (107/295) | | FEIRI (243/615) | | Mayo VDB (116/1141) | | DEFINE-FMD (108/126) | | UM-MGI/CCF (551/3436) | | Meta-analysis (1556 cases/7100 controls) | | | |
|---|---|---|---|---|---|---|---|---|---|---|---|---|---|---|---|---|---|---|---|
| | | | | OR | P | OR | P | OR | P | OR | P | OR | P | OR | P | OR (95%CI) | P | Direction | Het |
| rs9349379 (6:12903957) | PHACTR1 | A | 0.63 | 1.55 | $9 \times 10^{-7}$ | 1.33 | $1 \times 10^{-1}$ | 1.45 | $4 \times 10^{-2}$ | 1.32 | $6 \times 10^{-2}$ | 1.37 | $1 \times 10^{-1}$ | 1.43 | $3 \times 10^{-7}$ | 1.44 (1.31-1.57) | $5 \times 10^{-15}$ | ++++++ | 0.93 |
| rs11172113 (12:57527283) | LRP1 | T | 0.62 | 1.22 | $3 \times 10^{-2}$ | 1.07 | $7 \times 10^{-1}$ | 1.46 | $2 \times 10^{-1}$ | 1.62 | $2 \times 10^{-3}$ | 1.30 | $2 \times 10^{-1}$ | 1.40 | $8 \times 10^{-7}$ | 1.34 (1.22-1.46) | $2 \times 10^{-10}$ | ++++++ | 0.39 |
| rs7301566 (12:50581647) | LIMA1 | T | 0.45 | 1.16 | $7 \times 10^{-2}$ | 1.52 | $2 \times 10^{-2}$ | 1.09 | $6 \times 10^{-1}$ | 1.32 | $6 \times 10^{-2}$ | 1.55 | $2 \times 10^{-2}$ | 1.35 | $3 \times 10^{-6}$ | 1.29 (1.19-1.41) | $4 \times 10^{-9}$ | ++++++ | 0.39 |
| rs2681492 (12:90013089) | ATP2B1 | T | 0.84 | 1.42 | $2 \times 10^{-3}$ | 1.29 | $3 \times 10^{-1}$ | 1.45 | $9 \times 10^{-2}$ | 1.23 | $3 \times 10^{-1}$ | 1.42 | $2 \times 10^{-1}$ | 1.51 | $2 \times 10^{-5}$ | 1.43 (1.26-162) | $2 \times 10^{-8}$ | ++++++ | 0.96 |

Chr chromosome, pos position on build H19, EA effect allele, EAF effect allele frequency, OR odds ratio for genetic association in each cohort (logistic regression), P: P value of a two-sided Wald test, OR§: meta-analysis odds ratio (inverse variance-weighted method) Het.: P value for heterogeneity test between cohorts using Cochran's Q statistics.

found that associated variants from FMD GWAS (lead SNPs with $P < 10^{-4}$, and their proxies at $r^2 \geq 0.7$, Supplementary Data 1) were enriched in open chromatin peaks of artery tissues (average 1.5-fold enrichment, $10^{-2} < P < 7 \times 10^{-2}$, Supplementary Fig. S5d). To account for a potential effect of the number of peaks on SNP enrichment, which is lower in the arterial samples we analysed (Supplementary Fig. S5c), we restricted the analysis to the top scoring 50,000 peaks in SMCs, ECs, dermal fibroblasts and arteries, which confirmed the initial finding (Supplementary Fig. S5d). Globally, we found that artery-specific ATAC-Seq peaks are overrepresented in the vicinity of genes involved in contractile fibres and muscle system processes (Supplementary Fig. S5e, f).

We annotated the credible sets of variants, those representing >95% of cumulated posterior probability to cause the association at each locus, with overlapping open chromatin regions from these ATAC-Seq datasets (Table 3). At the *PHACTR1* locus, the lead variant (rs9349379) was located in a chromatin region open in arterial tissue but not accessible in VSMCs, fibroblasts, or ECs. This variant also overlapped with well-defined enhancer marks in arterial tissue and was strongly supported as the causal variant in this locus (Fig. 3a, b). The lead variant (rs11172113) in *LRP1* overlapped with open chromatin peaks in arterial tissue, primary VSMCs and fibroblasts, but not primary ECs (Fig. 3c, d). Strong enhancer and promoter marks were also present in arterial tissue around this variant, which mapped 5 kb downstream of *LRP1* promoter and is the most likely causal variant. At the *LIMA1* locus, the variant with the highest posterior probability of being causal, rs7301566, overlapped a region active in arteries, VSMCs, and fibroblasts and also strong enhancer marks in arteries (Fig. 3f). Three other variants (rs4459386, rs10783342, rs12815871) also overlapped marks indicating potential regulatory properties (Fig. 3f, Table 3). Finally, the *ATP2B1* locus included three highly associated and potentially causal variants (rs11105352, rs11105353 and rs11105354, Fig. 3g), which all overlapped a small open chromatin peak specifically observed in arterial tissue, and an enhancer-specific H3K4me1 histone mark (Fig. 3h).

**FMD potential target genes are expressed in VSMCs and fibroblasts**. We assessed the expression of FMD potential target genes (Table 2) by leveraging results from pre-existing transcriptomics datasets. Inquiry of the mouse aorta single-cell RNA-Seq data[20] showed that potential target genes were mostly expressed in VSMC and fibroblasts clusters. (Supplementary Fig. S6). Based on RNA-Seq dataset from GTEx, we found that FMD potential target genes were robustly expressed in tibial artery tissue, while *SLC24A3* presented a lower expression in arterial samples of women, compared to men ($P = 1 \times 10^{-5}$, Supplementary Fig. S7).

**FMD risk loci and associations with blood pressure and hypertension**. All FMD top loci were previously associated with at least one trait related to blood pressure, with the same alleles being associated with increased FMD risk and higher blood pressure/hypertension risk (Fig. 4a). Hypertension is reported in a large proportion of patients with FMD in general, and in the majority of FMD cases in this study (51–85%, Supplementary Table S1). To test for hypertension as a potential confounder driving the observed association signals, we conducted both stratified and adjusted analyses for hypertension status for the GWAS lead variants in the two largest studies: the French and the UM-MGI/CCF studies (Supplementary Table S6). Adjustment for hypertension status marginally modified the effects sizes and level of significance of the associations with FMD at all four loci, including when the association was absent (*LIMA1* in the French

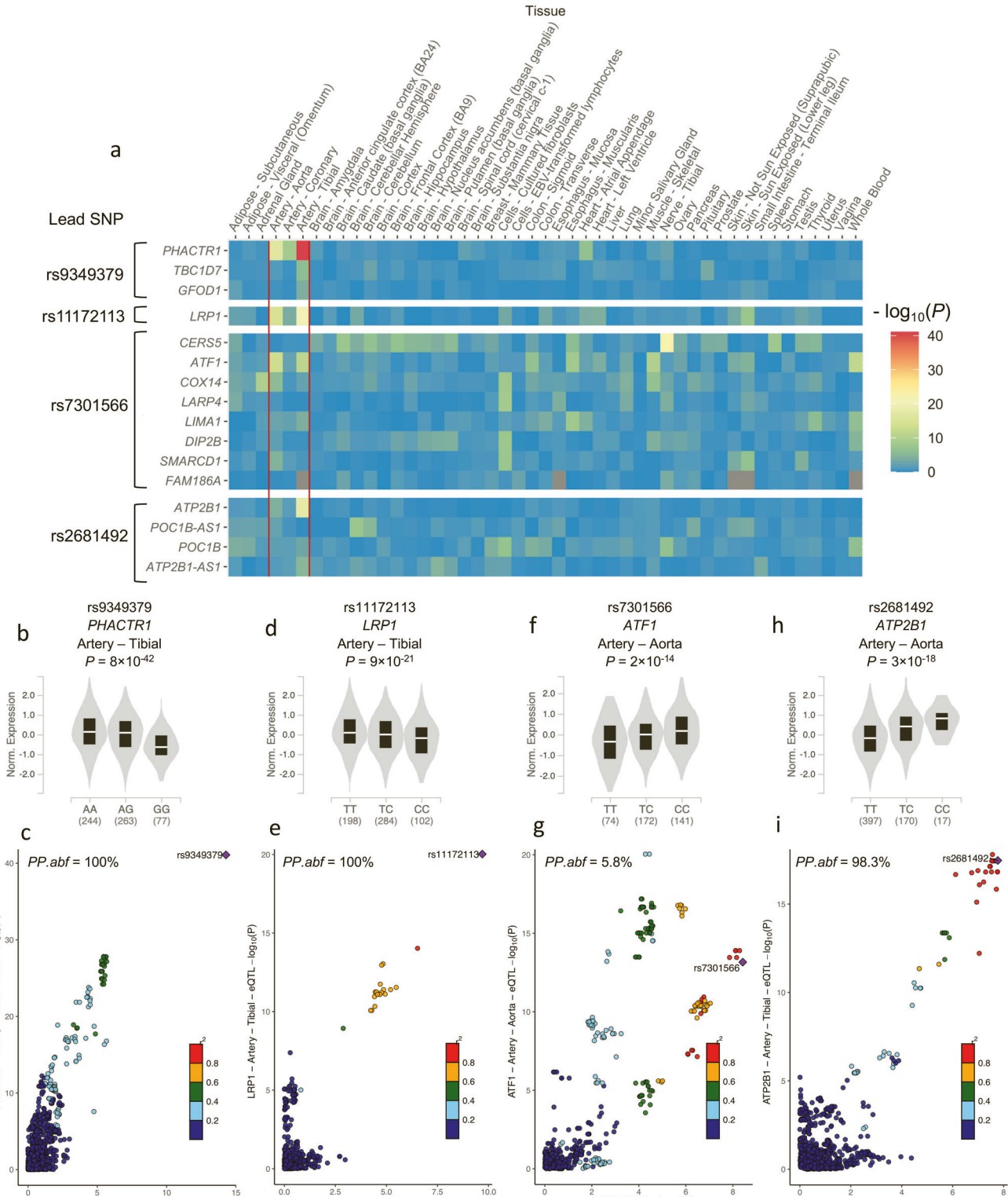

**Fig. 2 Tissue-wide eQTL signals near FMD loci. a** Heatmap representation of eQTL signals at FMD loci. GTEx v8 database was queried for significant eQTLs using the lead variant rsID number. All genes identified as positive eQTL in at least one tissue were selected to calculate eQTL associations in all tissues available in this database. The negative logarithms ($-\log_{10}$) of the eQTL association $P$ values are represented in a blue–red colour scale. **b, d, f, h** Violin plots representing normalized expression of *PHACTR1* (**b**), *LRP1* (**d**), *ATF1* (**f**) and *ATP2B1* (**h**) by genotype of lead SNPs in tibial artery (**b, d, h**) and aortic tissue (**f**). Plots illustrate the best eQTL association in arterial tissue for the lead SNP at each locus. eQTL unadjusted $P$ values were obtained from testing that the slope of linear regression model between genotype and expression deviates from 0 using a two-sided Wald test. FMD risk allele is on the left. **c, e, g, i** Colocalization plot of FMD association (x-axis, log scale of genetic association $P$ value) with tibial artery (**c, e, i**) and aortic tissue (**g**) eQTL association (y-axis, log scale of $P$ value) at each locus. Dot colour represents the LD $r^2$ with the lead variant in 1000G European samples. FMD lead variant is highlighted (Diamond shape, purple). Approximate Bayes Factor Posterior Probability (*PP.abf*) for the two traits to share a common causal variant (H4 coefficient) is indicated.

**Table 2 Transcriptome-wide association analysis in arterial tissues.**

| Locus information | | | | | Gene information | | TWAS | | | |
|---|---|---|---|---|---|---|---|---|---|---|
| Lead SNP | EA | EAF | OR | P | Gene | Chr: start-top | Best tissue | P | $P_{adj}$ | Z |
| rs9349379 | A | 0.63 | 1.44 (1.31–1.57) | $5 \times 10^{-15}$ | PHACTR1 | 6:12717893-13288645 | Aorta | $3 \times 10^{-13}$ | $4 \times 10^{-9}$ | 7.3 |
| rs11172113 | T | 0.62 | 1.34 (1.22–1.46) | $2 \times 10^{-10}$ | LRP1 | 12:57522276-57607134 | Tibial | $2 \times 10^{-10}$ | $3 \times 10^{-6}$ | 6.4 |
| rs2424245 | T | 0.86 | 1.37 (1.19–1.57) | $4 \times 10^{-6}$ | SLC24A3 | 20:19193290-19703581 | Tibial | $6 \times 10^{-7}$ | 0.01 | −5.0 |

The table P values (P) and Z scores (Z) obtained in the trancriptome-wide association based on gene expression models from GTEx in tibial artery, aorta and coronary artery and GWAS. For each gene, only the best TWAS association is shown. All genes with Bonferroni adjusted P value below 0.05 are reported. FMD GWAS association for the best SNP in the locus is indicated.
EA effect allele, EAF effect allele frequency, OR odds ratio, P: P value, Chr Chromosome, $P_{adj}$. Bonferroni corrected P-value for TWAS association.

study). Given the large proportion of hypertension, both in the cases and the controls of the French study, associations with FMD were only observed in the larger stratum of hypertensive patients. However, all four loci showed significant association with FMD both in hypertensive and non-hypertensive individuals in the UM-MGI/CCF case control study. We also looked up the lead variants significantly associated with systolic blood pressure (SBP) in a recent meta-analysis GWAS of blood pressure traits[21]. In addition to the three FMD loci (PHACTR1, LIMA1 and ATP2B1), one variant near FHL5 on chromosome 6 showed evidence for suggestive association with FMD ($P = 2 \times 10^{-5}$, Supplementary Data 2).

**FMD-associated loci are pleiotropically associated with multiple vascular diseases.** In addition to blood pressure, PHACTR1, LRP1 and SLC24A3 have been previously associated with migraine, involving identical risk alleles (Fig. 4a)[22]. PHACTR1 and LRP1 were also associated with cervical artery dissection and spontaneous coronary artery dissection, and LRPI was additionally associated with abdominal aortic aneurysm (Fig. 4a)[23–25]. PHACTR1 and ATP2B1 were associated with CAD and MI, while LRP1 was suggestively associated to CAD, and in all three loci, the CAD risk alleles were the opposite of those associated with FMD (Fig. 4a)[26]. Colocalization of FMD association signals with the most relevant traits (Fig. 4b, Supplementary Table S7), in addition to the comparison of FMD TWAS in arteries with other diseases (Fig. 4c), suggested that the same variants are likely to be causal in FMD, cervical artery dissection, migraine, and CAD.

**Genetic relationships between FMD and more common cardiovascular diseases and traits.** In light of the important number of diseases in which FMD loci and genes are involved, we used LD score regression[27] to calculate the genome-wide correlation between FMD and other diseases and traits (Fig. 5a, b, Supplementary Table S8). As expected, FMD was positively correlated with hypertension ($r_g = 0.36$, $P = 2 \times 10^{-7}$), systolic ($r_g = 0.43$, $P = 2 \times 10^{-9}$) and diastolic ($r_g = 0.37$, $P = 1 \times 10^{-8}$) blood pressure, and pulse pressure ($r_g = 0.35$, $P = 2 \times 10^{-8}$). FMD also correlated positively with migraine ($r_g = 0.28$, $P = 8 \times 10^{-4}$), intracranial aneurysm (pooled ruptured and unruptured, $r_g = 0.36$, $P = 2 \times 10^{-5}$), aneurysmal subarachnoid haemorrhage ($r_g = 0.35$, $P = 2 \times 10^{-4}$), and cervical artery dissection, although this latter did not survive correction for multiple testing ($r_g = 0.67$, $P = 3 \times 10^{-2}$). No significant correlation with any of the kidney function related traits was observed (Fig. 5b, Supplementary Table S8). To determine whether the correlation of FMD with blood pressure, an important risk factor for several CVDs, may influence the genetic correlations we estimated with FMD, we conditioned FMD association on the genetic association with SBP obtained from the meta-analysis GWAS[21] using mtCOJO. Of

note, the association of top loci with FMD remained significant after this conditioning on SBP genetics (Supplementary Table S9). Interestingly, mtCOJO results revealed a significant negative genetic correlation of FMD with CAD ($r_g = -0.31$, $P = 5 \times 10^{-5}$) and MI ($r_g = -0.30$, $P = 4 \times 10^{-4}$) (Fig. 5c, Supplementary Table S10), indicating that this opposite genetic relation between FMD and CAD/MI is not mediated by common loci between FMD and SBP. The genetic correlation of FMD with migraine was only marginally affected by conditioning on SBP genetics ($r_g = 0.33$, $P = 4 \times 10^{-4}$). On the other hand, the genetic correlations with intracranial aneurysm, subarachnoid haemorrhage, and low-density lipoprotein cholesterol levels were less significant after conditioning on SBP genetics, indicating that these correlations with FMD are in part due to genetic associated loci shared between FMD and SBP (Fig. 5c, d, Supplementary Table S10). The genetic correlation results were unchanged when we conducted the analyses after removing the SNPs at the top FMD-associated loci (Supplementary Tables S11–12).

**Discussion**
Our study is the most comprehensive genetic investigation to date dedicated to FMD, a non-atherosclerotic arterial disease primarily diagnosed in middle-aged women with few classical cardiovascular risk factors. Key findings from our study include: (i) identification of risk loci and/or genes using single variant GWAS and TWAS in arteries; (ii) through integration of newly generated and publicly available annotation datasets we narrowed down the list of causal variants in most risk loci; (iii) FMD risk genes showed consistent expression in arterial tissue, mostly in VSMCs and fibroblasts. (iv) We found an important genetic overlap between FMD and blood pressure, not only for the top associated loci and genes but also at the GWAS level, although hypertension is unlikely to confound the genetic association with FMD at the 4 loci described. (v) FMD shared genetic bases and potentially biological mechanisms with some but not all cardiovascular and neurovascular diseases.

The genetic investigation of FMD has been inconclusive for decades due to the lack of a clear genetic model. Here we provide an estimated SNP heritability of ~0.43 in its multifocal form, suggesting a polygenic basis of this neglected and underdiagnosed vascular disease. Efforts to establish large cohorts of FMD patients to conduct well-powered genetic studies are very recent and follow the increased awareness about its relatively high prevalence (~3%) in asymptomatic individuals[28].

Our study identified four loci contributing to FMD genetic risk. Most of these loci were previously shown to be involved in multiple vascular diseases. LRP1 is a risk locus for pulse pressure[29], migraine[22], aortic abdominal aneurysm[30], and was recently reported for spontaneous coronary artery dissection[24,31] involving the same risk allele for FMD that correlates with higher gene expression. However, the opposite allele was reported to

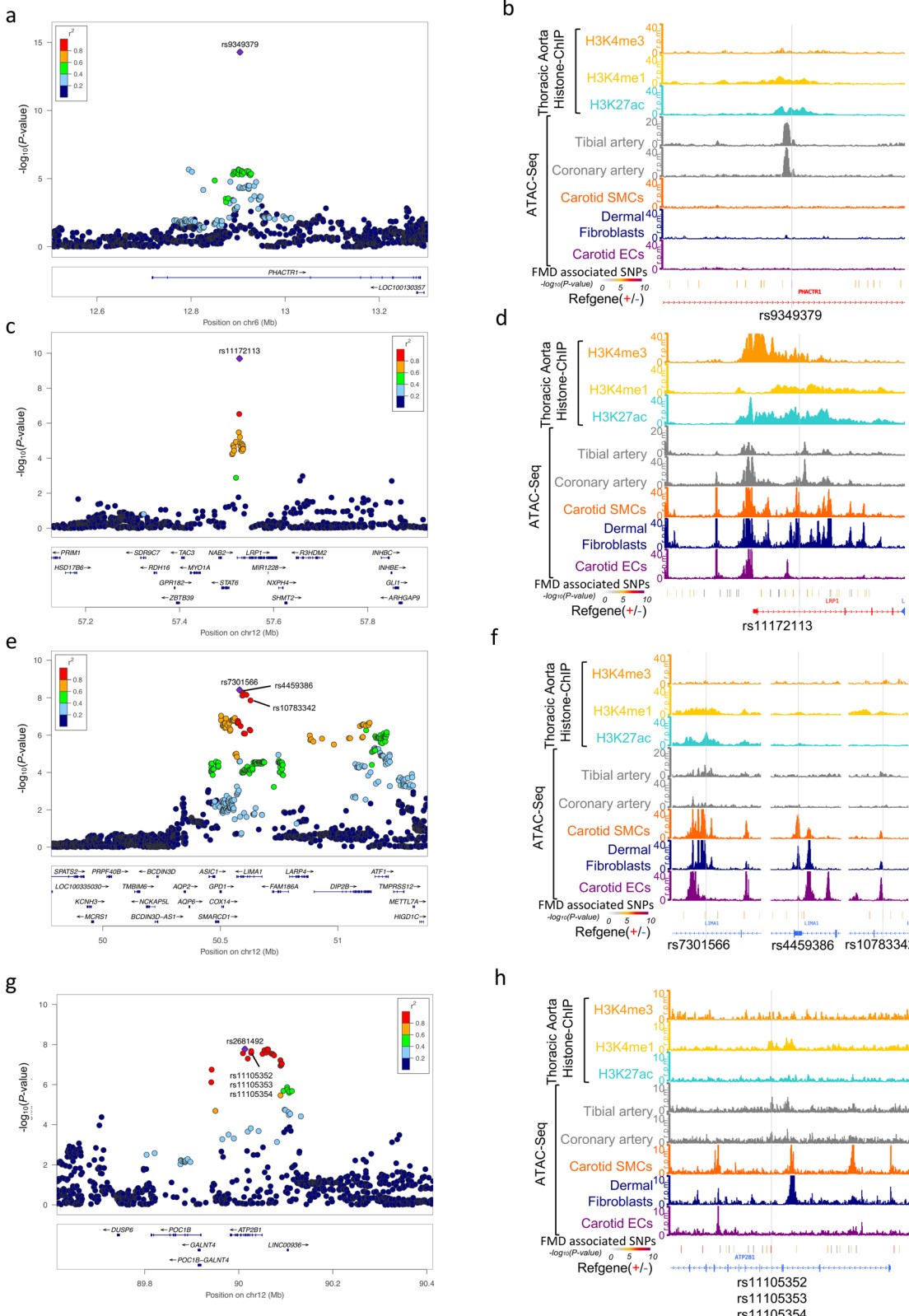

**Fig. 3 Visualization of potential causal variants genes at FMD-associated loci. a, c, e, g** LocusZoom representation of FMD-associated loci (**a** *PHACTR1* locus, **c** *LRP1* locus, **e** *LIMA1* locus, **g** *ATP2B1* locus). Dot colour indicates LD of each variant with the highlighted lead variant (purple diamond). Position and rsID of putative causal variants are indicated. *P* value of genetic association is represented on the *y*-axis (negative log scale) (**b, d, f, h**) Genome browser visualization of ATAC-Seq/Histone-ChIP read densities (in reads/million, r.p.m.) in the regions surrounding putative causal variants. **b** *PHACTR1* locus. **d** *LRP1* locus. **f** *LIMA1* locus. **h** *ATP2B1* locus. Grey line highlights variant position.

**Table 3 Candidate causal variants at FMD-associated loci.**

| Locus | SNP information | | | | | | ATAC-Seq | | | | | | Histone marks | | |
| | | | | | | | Coronary artery | | | Carotid artery | | Skin | Any arteries | | |
| Lead SNP | Closest gene | N SNPs | SNP | Chr:pos (hg19) | P | Post. Prob. | Whole tissue | SMC | EC | SMC | EC | HDF | H3K4me1 | H3K4me3 | H3K27ac |
|---|---|---|---|---|---|---|---|---|---|---|---|---|---|---|---|
| rs9349379 | PHACTR1 | 1 | rs9349379 | 6:12903957 | $5 \times 10^{-15}$ | 100.0% | + | | | | | | + | | + |
| rs11172113 | LRP1 | 1 | rs11172113 | 12:57527283 | $2 \times 10^{-10}$ | 99.9% | + | + | | + | | | + | + | + |
| rs7301566 | LIMA1 | 24 | rs7301566 | 12:50581647 | $4 \times 10^{-9}$ | 23.5% | | + | | + | | + | + | | + |
| | | | rs4459386 | 12:50594947 | $5 \times 10^{-9}$ | 17.3% | | + | | + | | + | + | | |
| | | | rs10783342 | 12:50628466 | $1 \times 10^{-8}$ | 6.7% | | | | | + | | + | | |
| | | | rs12815871 | 12:50550948 | $2 \times 10^{-7}$ | 0.6% | | | | + | | + | + | | |
| rs2681492 | ATP2B1 | 18 | rs11105354 | 12:90026523 | $2 \times 10^{-8}$ | 6.9% | + | | | | | | + | | |
| | | | rs11105352 | 12:90026462 | $3 \times 10^{-8}$ | 5.6% | + | | | | | | + | | |
| | | | rs11105353 | 12:90026463 | $3 \times 10^{-8}$ | 5.6% | + | | | | | | + | | |

All variants within the 95% credible set in the FMD association study were tested for overlap with open chromatin regions in carotid/coronary artery-derived primary cells and dermal fibroblasts described in this study, open chromatin in coronary artery tissue[21] and histone marks in artery tissues (aorta, coronary or tibial arteries) from ENCODE database. N SNPs indicate the number of SNPs tested at each locus. P indicates the GWAS association P value of the SNP in the reported association. Posterior probability of each variant for being a causal variant is indicated (Post. Prob.). + indicates overlap of the variant with at least one dataset of the corresponding category.

increase the risk for MoyaMoya disease[32] and suggestively for CAD[26]. LDL receptor protein 1 (LRP1) is a widely studied molecule that plays important roles in multiple cellular processes relevant to FMD such as the remodelling of the extracellular matrix and VSMCs migration[33], among others. LRP1 function in VSMCs is mediated partly by the modulation of calcium signalling, leading to deficient vasoconstriction in VSMC-specific Lrp1-deficient mice[34].

Interestingly, two of the genes we identified are involved in intracellular calcium homoeostasis, a highly relevant molecular mechanism for vascular contractility and vasodilation. ATP2B1 encodes an ATP-dependent calcium channel specialized in the exportation of calcium ions from the cytoplasm to the extracellular space. ATP2B1 is a well-established hypertension and blood pressure locus, where the same alleles increase the risk for FMD and hypertension (Fig. 4). Mice lacking Atp2b1 specifically in VSMCs exhibit hypertension, higher intracellular calcium levels and increased sensitivity to nicardipine, a calcium channel blocker[35,36]. On the other hand, SLC24A3, which we identified through the TWAS analyses in arteries, encodes a transmembrane sodium/potassium/calcium exchanger also involved in calcium homoeostasis[37]. The relevance of impaired vasodilation and/or enhanced vasoconstriction in FMD pathogenesis is supported by our recent study where we reported an enrichment among FMD patients for rare loss-of-function mutations in the gene encoding the receptor for prostacyclin, a major vasodilator hormone[38]. In line with these findings, impaired dilation of arteries in response to sublingual glyceryl trinitrate, a proxy for VSMC dysfunction, was reported in FMD patients, including in arterial segments clinically unaffected by the disease[39].

Several potential target genes were identified in the chromosome 12 LIMA1 locus, with LIM domain and acting binding 1 (LIMA1) and activating transcription factor 1 (ATF1) genes as the strongest biological candidates. The role of LIMA1 in actin dynamics is compatible with a potential role in maintaining cell shape, a feature lost in FMD-affected VSMCs[40]. A study described a rare frameshift variant in LIMA1 from a family with inherited low LDL cholesterol, and resistance in diet-induced hypercholesterolemia has been described in a Lima1-deficient mouse[41]. While its specific role in arteries is not known, ATF1 is a transcriptional effector of the cyclic adenosine monophosphate pathway, which plays a key role in the regulation of vascular tone[42].

Our study provides robust confirmation of the association with FMD of PHACTR1, a pleiotropic locus that is involved in a large number of vascular diseases[43,44]. The functional annotations, the colocalization analyses, TWAS in arteries and eQTL results, all point to rs9349379 as a clear regulator of PHACTR1. The previously suggested regulation of the endothelin-1 gene (EDN1)[43] is not supported by our results, or those from a subsequent study that used an identical approach of iPSC induced endothelial cells[45], or yet another study which directly measured endothelin-1 plasma levels in FMD patients and matched healthy controls[16]. The phosphatase and actin-binding protein encoded by PHACTR1 regulates actin stress fibre assembly and cell motility[46], functions highly relevant to the cellular disorganization that characterizes VSMCs in FMD-affected arteries[40]. Further investigation, especially with in vivo models, is needed to more precisely define the function of PHACTR1 in the context of the genetic risk to a diverse panel of CVDs.

Through genetic correlation analyses, we were able to globally position the genetic basis of FMD among the genetics of more commonly studied CVDs and traits. We showed that FMD shares a significant proportion of its genetic basis with hypertension and blood pressure-related traits, not only for the top associated loci and genes but also genome-wide. The look-up for

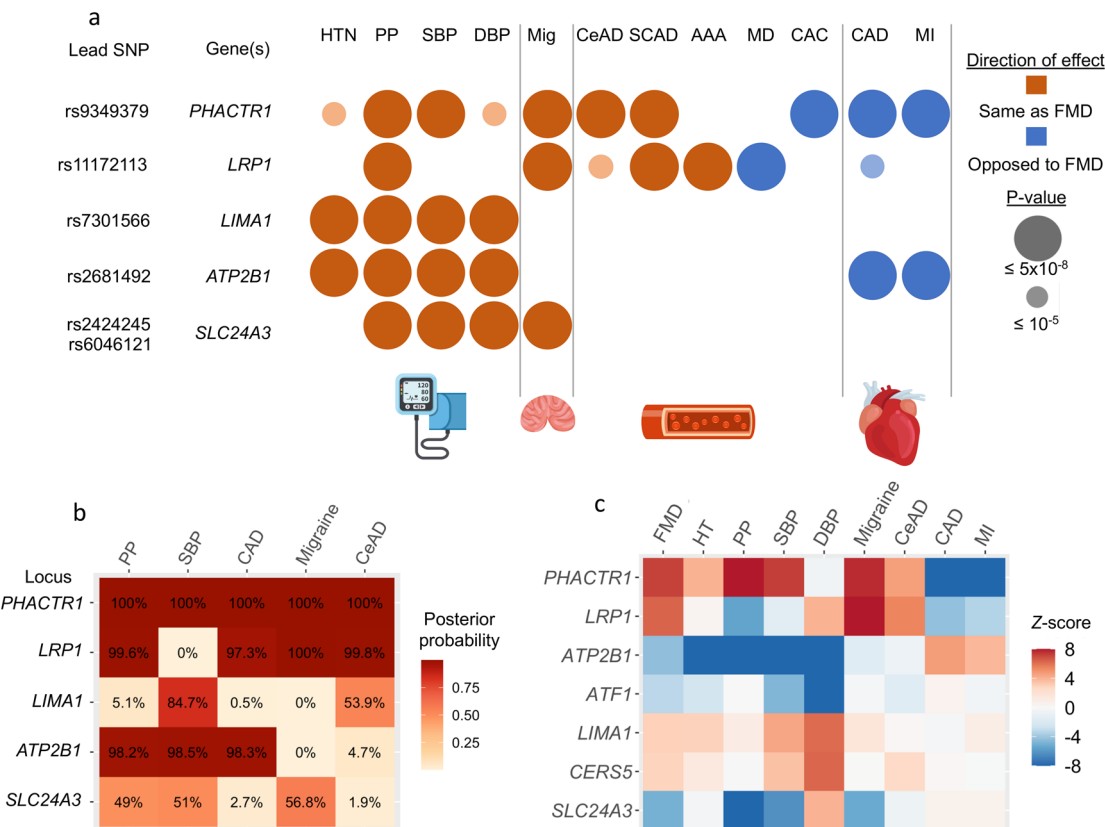

**Fig. 4 Pairwise trait colocalization of FMD associations. a** Associations of FMD loci with other vascular diseases. Variants in LD ($r^2 > 0.5$) with the lead SNP were used to query GWAS catalogue database (accessed on August 5th 2020), UK BioBank GWAS traits and specific meta-analysis of GWAS for CAD/MI, stroke, blood pressure and intracranial aneurysms. Two independent lead SNPs were retained for *SLC24A3* locus. Overlaps are reported for the following traits/diseases: hypertension (HTN), pulse pressure (PP), systolic blood pressure (SBP), diastolic blood pressure (DBP), migraine (Mig), cervical artery dissection (CeAD), spontaneous coronary artery dissection (SCAD), abdominal aortic aneurysm (AAA), MoyaMoya disease (MD), coronary artery calcification (CAC), coronary artery disease (CAD) and myocardial infarction (MI). Large bubbles indicate association below genome-wide significance for the corresponding trait (*P* value of genetic association $< 5 \times 10^{-8}$), smaller bubbles correspond to suggestive signals ($P < 1 \times 10^{-5}$). Red colour indicates same direction effects of risk alleles compared to the association with FMD, blue colour opposite direction associations. **b** Heatmap representation of the approximate Bayes factor posterior probability of FMD and related traits to share a common causal variant at the indicated loci (coefficient H4 of the colocalization analysis, represented on a white-red colour scale). **c** Heatmap representation of TWAS *Z*-Score for FMD-associated genes. TWAS was performed with tibial artery gene expression models for the indicated traits or diseases (*x*-axis). *Z*-scores, represented on a blue-red colour scale, are shown for all FMD-associated genes in the gene-based and TWAS analyses (*y*-axis). Illustrations were designed by macrovector/Freepik.

the currently large list of SBP loci in our meta-analysis of GWAS only revealed one additional suggestive locus near *FHL5*, potentially reflecting the limited power of our datasets to detect additional loci using the existing list of SBP loci. Although we have demonstrated that hypertension is not a confounding factor for the associations of FMD top loci, the exact direction of the relationship between the genetic basis of FMD and hypertension is challenging to determine given that we only report 4 loci, compared to >1000 SBP currently reported loci[21,29]. FMD often manifests with renovascular hypertension. Although percutaneous angioplasty leads to blood pressure normalization in ~40% of cases, it can also be ineffective, suggesting potential additional mechanisms leading to hypertension in individuals with FMD[47]. Not all patients with FMD present with hypertension, especially when the disease affects arteries outside the renal bed[8]. Another possible explanation of the genetic correlation with SBP could be the high prevalence of essential hypertension in the general population, which is a major risk factor for all forms of CVD where FMD can coexist, including stroke and MI. Further investigation on FMD aetiology will provide additional insights into the clinical and genetic relationship of FMD, SBP, and hypertension.

We showed an expected positive genetic correlation between FMD and migraine, which is reported by 25–69% of FMD patients[4,8], and cervical artery dissection, which occurs in the same cerebrovascular beds affected in FMD (i.e., carotid and vertebral arteries). However, we found little support for shared genetics with ischemic stroke subtypes despite FMD being frequently diagnosed in the context of a stroke event. On the other hand, FMD seems to be more genetically related to intracranial aneurysm and aneurysmal subarachnoid haemorrhage, concordant with the observation that approximately 12–13% of patients with FMD are found to have intracerebral aneurysms upon angiographic imaging surveillance[4]. Stroke due to subarachnoid haemorrhage shares several clinical characteristics with FMD, mainly a high proportion of younger (<55 years) and more women among patients, association with blood pressure and smoking[48]. Finally, we observed an inverse association between FMD and CAD/MI both for the top loci and genes, and globally when FMD association is conditioned on the genetics of SBP. The elimination of the presence of atherosclerosis as the cause of stenoses and aneurysms is required for FMD diagnosis, which may have influenced this negative correlation with a disease where atherosclerosis is underrepresented. FMD is a disease

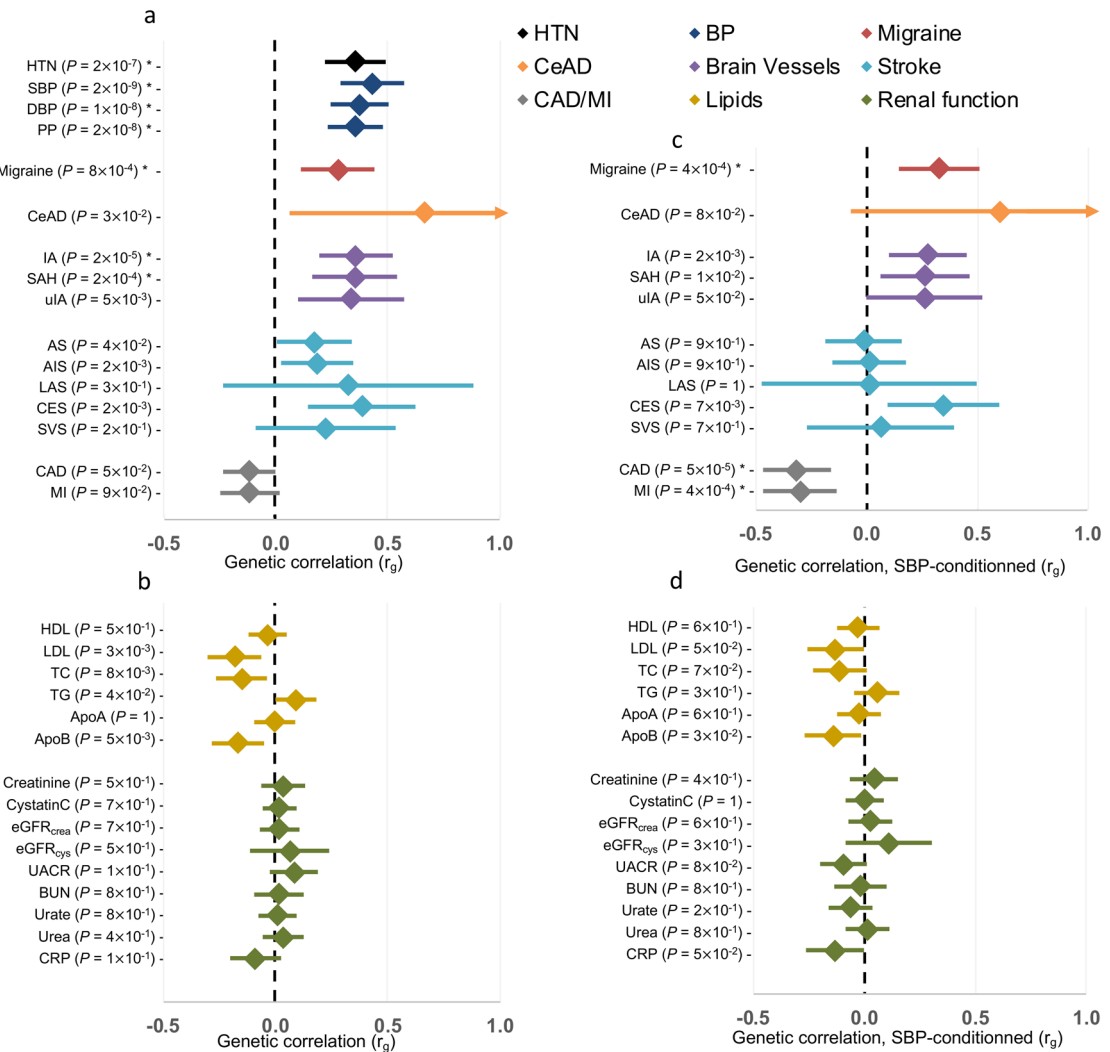

**Fig. 5 Genetic correlations between FMD and other traits and diseases. a** Genetic correlation obtained using LD Score analyses between FMD and vascular diseases and traits. HTN hypertension, BP blood pressure, SBP systolic BP (dark blue), DBP diastolic BP, PP pulse pressure, CeAD cervical artery dissection, IA intracranial aneurysm, all forms, SAH aneurysmal subarachnoid haemorrhage, i.e., ruptured intracranial aneurysm, uIA unruptured intracranial aneurysm, AS any stroke, AIS any ischaemic stroke, LAS large artery stroke, CES cardioembolic stroke, SVS small vessel stroke, CAD coronary artery disease, MI myocardial infarction. **b** Genetic correlation obtained using LD Score analyses between FMD and vascular diseases and metabolic traits. HDL high-density lipoprotein, LDL low-density lipoprotein, TC total cholesterol, TG triglycerides, ApoA apolipoprotein A, ApoB apolipoprotein B, eGFR estimated glomerular filtration rate (calculated with creatinine or cystatin), UACR urine albumin to creatinine ratio, BUN blood urea nitrogen, CRP C-reactive protein. **c**, **d** Genetic correlation results after FMD genetic association statistics were conditioned on systolic blood pressure. *Unadjusted $P$ value of genetic correlation estimated from linkage disequilibrium score regression using a two-sided Wald test is below the multiple testing threshold for significance ($1.6 \times 10^{-3}$ for 31 tests).

mostly diagnosed in young to middle-aged women, who are less prone to develop CAD and MI, which predominantly affects older men. Whether FMD patients are less likely to develop CAD/MI later in life, compared to patients of similar clinical characteristics is not known. Endogenous or exogenous female hormones are considered to be protective factors from CAD/MI[49], but are suspected as a potential risk factor in FMD pathogenesis, given the high proportion of women among patients (80–90%) and the age of diagnosis of FMD (on average ~50 years)[4]. Oestrogens stimulate the release of vasodilator mediators such as nitric oxide and prostacyclin and inhibit the potent vasoconstrictor endothelin-1[49]. Our study did not point to any direct link with sex hormone metabolism or regulation, except sex differences in the level of *SLC24A3* transcript expression in women as compared to men, consistent with the direction of effects of FMD risk allele. Interestingly, *Slc24a3* expression was previously

described as regulated during the oestrous cycle in rodents[50]. More knowledge about detailed biological and physiological roles of both genes are needed to address potential consequences on artery remodelling in sex and atherosclerosis dependent contexts.

Our work has several limitations. The study has limited power to fully address the genetic basis of FMD. This is in part due to underdiagnosis of FMD[4] and overall modest sample size. Future efforts to analyse larger cohorts are expected to permit further investigation of the relationship of FMD to other vascular diseases, including Mendelian Randomization approaches that would benefit from higher-powered analysis. Specifically, the current findings are insufficient to allow an extensive investigation of the relationship between FMD and hypertension to address putative causal relationships; the main limitations being the extremely unbalanced number of established loci that can be used as instruments (4 loci for FMD as compared to ~800 for

SBP), with all FMD loci established as having pleiotropic association with blood pressure and several other vascular diseases. Due to the need for angiographic imaging for the identification of FMD, it is possible that asymptomatic FMD was present in a small proportion of control subjects. We compared our genetic data to available expression datasets in tibial artery, coronary artery and aorta, but arterial beds most frequently affected by FMD lesions (renal or carotid artery) would be more appropriate. Finally, due to very low numbers of non-Europeans individuals in our studies, we focused our analysis on European ancestry subjects. The study of FMD in diverse ethnic groups will be required to analyse the trans-ethnic genetic basis of FMD.

In summary, in this first meta-analysis of GWAS for FMD, we report robustly associated loci and genes and provide several leads toward understanding biological mechanisms of arterial stenosis in the absence of atherosclerosis. Further investigation of the exact biological effects driven by these genes may shed light on the cause of the higher prevalence of FMD in women and provide insights into the shared genetic basis between FMD and more common CVDs and traits.

## Methods

**Patients and control populations**. The meta-analysis included participants of European ancestry from six studies: ARCADIA[8]/3C[51] GWAS, Mayo Vascular Disease Biorepository[52], DEFINE-FMD study[16], ARCADIA-POL[53]/WOBASZ II[54] study, University of Michigan/Cleveland Clinic Foundation (UM/CCF) study[31,55] and FEIRI[7]/ASKLEPIOS[56] study. FMD patients presented similar clinical characteristics (Supplementary Table S1) and homogeneous diagnosis, exclusion and inclusion criteria. Detailed description of the participating cohorts is given below.

**ARCADIA**. FMD patients were included from the ARCADIA (Assessment of Renal and Cervical Artery DysplasIA) register, a national FMD registry at the Georges Pompidou European Hospital, APHP, Paris[8]. The diagnosis of FMD was established using clinical information from the medical history, the interpretation of angiography and/or computed tomography scan of arterial beds after the exclusion of other causes of arterial stenosis such as atherosclerosis, Takayasu disease, Ehlers Danlos syndrome and neurofibromatosis type 1. Given the complexity of the interpretation of imaging of vascular diseases, a local panel of experts including clinicians from the departments of hypertension, radiology, vascular medicine and medical genetics validated the diagnosis. ARCADIA/PROFILE protocol was approved by the Ile-De-France research ethics committee (Comité de Protection des Personnes: CPP d'île de France) on 03/04/2009 (ID: 2009-A00288-49).

**Three City (3C)**. The Three-City Study (3C Study) is a population-based longitudinal study of the relation between vascular diseases and dementia in persons aged 65 years and older[51]. Participants were recruited from three French cities: Bordeaux (South-West), Dijon (North-East) and Montpellier (South-East). The 3C Study extended from 1999 to 2012. Participants underwent regular extensive examination. Examination included measurements of traditional vascular risk factors (blood pressure, glycaemia, lipids, etc.), cognitive functions, and subclinical vascular diseases using carotid ultrasound and cerebral magnetic resonance imaging. The study protocol was approved by comité consultatif de protection des personnes dans la recherche biomédicale Bicêtre Hôpital Bicêtre n°99-28 CCPPRB approved 10/06/99, 11/03/2003 and 17/03/2006.

**Mayo VDB Case Control study**. Potential FMD cases and controls were identified from the Mayo Vascular diseases Biorepository (VDB)[52]. Electronic health records of 209 patients including angiographic imaging (computed tomographic, magnetic resonance or catheter-based angiography) were manually reviewed by clinicians and the diagnosis was confirmed for 175 patients according to previously reported criteria[4], of whom 169 (83.4% females) had high-density genotyping data following QC. Controls ($n = 1141$) were participants without known FMD or atherosclerotic vascular diseases. The study was approved by the Mayo Clinic Institutional Review Boards (IRB # 08-008355).

**DEFINE-FMD Study**. Eligible cases were females with an imaging-confirmed diagnosis of multifocal FMD and who fulfilled other accepted diagnostic criteria[4,16]. DEFINE-FMD purpose was to establish a library of fibroblasts, DNA, plasma and serum from patients with FMD and unaffected healthy control subjects. FMD cases were required to have a clinical diagnosis of multifocal FMD that is confirmed by imaging [computed tomographic angiography (CTA), magnetic resonance angiography, or catheter-based angiography]. Healthy controls were matched to FMD cases according to age and sex, required to be receiving ≤2 blood pressure medications, have a body mass index <35 kg/m² and to be non-smokers.

Healthy controls underwent physical exam and those with bruits; unexplained hypertension or other cardiovascular findings were excluded. Exclusion criteria for all subjects included male gender, unifocal FMD, use of immunosuppressive agents, major comorbidities, diseases that may confound genetic/genomic analyses (i.e., Crohn's disease, multiple sclerosis, etc.) or any other form of heritable vascular disease (i.e., Ehlers-Danlos, Marfan, Loeys-Dietz)[16]. DEFINE-FMD study was approved by the Human Research Ethics Committee of the Icahn School of Medicine at Mount Sinai (Study ID: HS#13-00575/GCO#13-1118) and is registered with ClinicalTrials.gov Identifier: NCT01967511. Patient consent did not involve the possibility to redistribute biological material and any request should be addressed to DEFINE-FMD leaders.

**ARCADIA-POL**. Patients were recruited through the ARCADIA-POL (Assessment of Renal and Cervical Artery Dysplasia - POLAND) study, a nation-wide Polish registry for FMD[53]. From January 2015 until December 2018, 343 patients, women and men aged more than 18 years were referred from 32 centres in Poland and were evaluated for suspicion of FMD in the Institute of Cardiology, Warsaw, Poland. FMD lesions in at least one vascular bed were confirmed in 232 patients using whole-body angio-computed tomography. We analysed 129 patients for whom we excluded atherosclerotic stenosis and syndromes where FMD-like lesions are often observed (eg, Ehlers Danlos, Loeys-Dietz). ARCADIA-Pol study was approved by Local Ethics Comittee, Institute of Cardiology, IK-NPIA-0021017/1482/17.

**WOBASZ II**. We used 298 controls from a randomly ascertained sample from the WOBASZ II (Multicentre National Population Health Examination Survey) study, a population-based Polish cohort[54]. The WOBASZ II study was planned as a cross-sectional survey of a random sample of Polish residents aged over 20 years. The selection, using the National Identity Card Registry of the Ministry of Internal Affairs, was made as a three-stage sampling, stratified according to administrative units (voivodeships), type of urbanisation (commune), and gender. The study protocol consisted of a questionnaire used in face-to-face interviews, physical examination, and blood samples. WOBASZ II was coordinated by the Department of Epidemiology, Cardiovascular Diseases Prevention and Health Promotion of the Institute of Cardiology in Warsaw in cooperation with medical universities in Gdansk, Katowice, Krakow, Lodz, and Poznan. The study was accepted by the Field Bioethics Committee of the Institute of Cardiology in Warsaw.

**University of Michigan-Michigan Genomics Initiative/Cleveland Clinic case control study**. FMD patients were recruited at the UM and/or the Cleveland Clinic. UM/CCF FMD cases were recruited into an IRB-approved study through a referral clinic at the UM and through self-referral to the study. The Cleveland Clinic cases were enroled among consecutive patients seen at a dedicated FMD referral clinic. Clinical diagnosis of FMD was ascertained by a vascular medicine specialist after review of diagnostic imaging (computed tomographic, magnetic resonance or catheter-based angiography) and prior to blood sample collection. Genomic DNA was isolated from a peripheral blood or saliva sample. The Michigan Genomics Initiative (MGI) is a program that recruited participants while awaiting diagnostic, interventional, and surgical procedures. Participants provided a blood sample for genetic analysis and agreed to link their sample to their electronic health record and other sources of health information[55]. Several ICD codes corresponding to diagnoses of arterial aneurysm, dissection, and non-atherosclerotic dysplasia and stenosis were excluded, as well as connective tissue disorders, as previously described[31]. The hypertension variable was defined as any ICD code containing the "hypertension" term for MGI controls. The UM/Cleveland case control study was approved by each institution's IRB protocols: The University of Michigan IRB number is HUM00044507, and the Cleveland Clinic IRB number is 10-318.

**FMD European/International Registry and Initiative (FEIRI)**. Patients were enroled through the European/International FMD Registry and Initiative (FEIRI)[7]. FMD was defined as the presence of an idiopathic, segmental, non-atherosclerotic, and noninflammatory stenosis (either with focal or string-of-beads appearance) of a small- or medium-sized artery in at least 1 vascular bed, documented with CTA, magnetic resonance angiography, or catheter-based angiography imaging. Patients with a suspicion of FMD only based on duplex ultrasound were excluded. Patients whose primary diagnosis was spontaneous coronary artery dissection were not eligible, even in the presence of extracoronary FMD lesions. Data on demographic and FMD characteristics were collected through the FMD Registry. All centres included in FEIRI received approval from the respective local/ national ethics committees.

**Asklepios**. The Asklepios Study is a longitudinal population study focusing on the interplay between ageing, cardiovascular haemodynamics and inflammation in preclinical CVD[56]. Participants aged 35–55 years were sampled from the twinned Belgian communities of Erpe–Mere and Nieuwerkerken. Exclusion criteria were the presence of atherosclerotic lesions and concomitance with major illness, diabetes or conditions precluding accurate haemodynamic assessment. All participants underwent systematic physical examination during a continuous 2-year

period, between October 2002 and September 2004 at a single study site in Erpe–Mere involving: measurement of basic clinical data, blood samples collection, echocardiographic examination and vascular echography and tonometry measurements. The ethical committee of the Ghent University Hospital approved the study protocol.

**Genome-wide association analyses and meta-analysis**. Details on genotyping, variant calling for each cohort and pre-imputation quality control in each study are listed in Supplementary Table S13. In brief, genotyping was performed using commercially available arrays. To increase the number of tested SNPs and the overlap of variants available for analysis between different arrays, all European ancestry cohorts imputed genotypes to the most current HRC v1.1 reference panel[57] on the Michigan Imputation Server[58]. GWAS was conducted in each study under an additive genetic model using PLINK v2.0[59]. For chromosome X, males and females were both on a 0..2 scale (chromosome X inactivation assumption model). Models were adjusted for population structure using the first five principal components, sex (except in the women-only analyses) and study-specific genomic control. Prior to meta-analysis, we removed SNPs with low minor allele frequencies (MAF) (<0.01), low imputation quality ($r^2 < 0.8$), and deviations from Hardy–Weinberg equilibrium ($P < 10^{-5}$). A total of 5,483,710 variants met these criteria and were kept in the final results.

Association results were combined using an inverse variance-weighted fixed-effects meta-analysis in METAL software[60], with correction for genomic control. Heterogeneity was assessed using the $I^2$ metric from the complete study-level meta-analysis. Between-study heterogeneity was tested using the Cochran Q statistic and considered significant at $P \leq 10^{-3}$ Genome-wide significance threshold was set at the level of $P = 5.0 \times 10^{-8}$. LocusZoom (http://locuszoom.org/) was used to provide regional visualization of results. Statistical analyses and plotting were performed using R (3.6.1), through RStudio interface software (v1.2.335). Following packages were used: MatrixEQTL_v.2.1.1, LDlinkR_1.0.2, ggrepel_0.8.2, locuscomparer_1.0.0, coloc_3.2-1, dplyr_1.0.2, tidyr_1.1.1, RColorBrewer_1.1-2, ggplot2_3.3.3.

**eQTL and colocalization analyses**. We queried the GTEx database (v8 release)[61] with rsID of lead variants at FMD loci in arterial tissues for major associated genes (permutations q value < 0.05). For each identified gene, variant-gene association was queried in all tissues using "eQTL calculator" function on GTEx website (https://gtexportal.org/home/). Uncharacterized non-coding transcripts were excluded from the analysis. For colocalization, all SNP Gene associations in the 3 arterial tissues were retrieved from GTEx to compare with the FMD GWAS meta-analysis result and the multifocal FMD GWAS meta-analysis result. We generated colocalization plots using locuscompareR package[62] and Bayesian posterior probability was calculated using coloc.abf function in R coloc package[63]. The H4 coefficient indicating the probability of the two traits to share a causal variant was reported.

**Gene-based and transcriptome-wide association analyses**. Gene-based association was conducted using the MAGMA tool[13], implemented in the FUMA platform[14]. Locations of protein-coding genes were defined as the regions from transcription start site to transcription stop site (default option in MAGMA). TWAS was performed using FUSION R/python package[17]. Gene expression models were precomputed from GTEx data (v7 release) and were provided by the authors. Only genes with heritability $P$-value < 0.01 were used in the analysis. Uncharacterized non-coding transcripts were excluded from the analysis. Both tools used linkage disequilibrium information from the European panel of the 1000 Genomes phase 3. Bonferroni multiple testing correction was applied using the p.adjust function in R (v 3.6.1).

**Primary cell culture and ATAC-Seq experiments**. With the exception of the dermal fibroblast cell lines obtained under the DEFINE-FMD protocol, other primary cells were purchased from Cell Applications (San Diego, CA) except HDF (ATCC, Manassas, VA) and cultured with 5% $CO_2$ in a 37 °C incubator following manufacturer's instructions. VSMC cells were grown in DMEM supplemented with 5% FBS, Insulin (5 µg/mL), EGF (0.5 ng/mL), bFGF (2 ng/mL) and antibiotics (all from Thermo Fisher Scientific, Waltham, MA, USA). Cells were at passage 5 (HDF, HCF, HCtAEC, HCAEC) or 6 (HCtASMC, HCASMC) for ATAC-Seq analyses.

**Expression quantitative trait loci analysis on fibroblasts from FMD patients**. Fibroblasts of the DEFINE-FMD Study were derived from skin biopsy samples using standard explant techniques from 83 FMD patients participating to the genetic study as previously described[3]. Briefly, skin biopsy samples were dissected into small pieces and cultured under cover slips in DMEM/F12 medium containing 20% foetal bovine serum, 1% antibiotic-antimycotic solution, 1% 200 mM L-glutamine, 1% 100 mM Sodium pyruvate, and 1% MEM Non-Essential Amino Acids (all from Life Technologies, Grand Island, NY, USA) at 37 °C in 5% $CO_2$. Cell culture medium was replaced every 48 h and confluent cells from passages 2–3 were used for RNA extractions.

Whole transcriptome sequencing was performed on an Illumina HiSeq2000 using single end following 100 base pair library preparation. Quality control was performed using FASTQC (www.bioinformatics.babraham.ac.uk/projects/fastqc/) that checks raw sequence data for per-base quality, per-sequence quality, number of duplicate reads, number of reads with an adaptor, sequence length distribution, per-base GC content, per-sequence GC content and Kmer content. GENCODE 29 was used as reference annotation to quantify gene and long non-coding RNA (lncRNA) expression. Sequencing reads (fastq files) were mapped to GRCh38 human reference genome using STAR aligner (version 3.6.0c) with default mapping parameters[64]. Low counts were removed by keeping genes where the count per million is greater than 1 in at least two samples. After filtering zero and low counts, an average of 34 million reads was retained per sample.

R package Matrix eQTL v.2.1.1 was used to compute *cis*-regulated eQTLs in the cohorts of cases, controls and cases plus controls, respectively[65]. Age and the first five genetics principal components were applied as covariates. Case control status was accounted as a covariate effect for the case control pooled cohort. All cis-regulatory SNPs were located within 1 Mb of the gene by using the hg19 genome (Ensembl Gene annotation build 75) and statistics were calculated by a linear model.

**Assay for transposase accessible chromatin (ATAC-Seq)**. Primary cells at passage 5 or 6 were trypsinized, counted using a Nucleocounter NC-100 (Chemometec, Allerod, Denmark), and 50,000 cells were collected for each experiment. ATAC library preparation was performed as described in the Omni-ATAC protocol[66,67]. Briefly, cells were washed in PBS and incubated for 3 min in 50 µL Resuspension Buffer (10 mM Tris-HCL, pH 7.4, 10 mM NaCl, 3 mM $MgCl_2$, all from Sigma-Aldrich, Saint-Louis, MI, USA) containing 0.1% Tween 20 (Sigma-Aldrich, Saint-Louis, MI, USA), 0.1% IGEPAL CA-630 (Sigma-Aldrich, Saint-Louis, MI, USA) and 0.01% Digitonin (Thermo Fisher Scientific, Waltham, MA, USA), before 1 mL Resuspension Buffer with 0.1%Tween 20 was added to the cells. After centrifugation, isolated nuclei were resuspended in Transposition Buffer (50 µL, made from 16.5 µL PBS, 5 µL $H_2O$, 0.5 µL Tween 20 (10%), 0.5 µL Digitonin 1%, 25 µL 2× TD Buffer and 2.5 µL TD Enzyme, last two from Illumina, San Diego, CA, USA). Then, the transposition reaction was purified with MinElute PCR Purification Kit (QIAGEN, Hilden, Germany). Transposed DNA was preamplified for 5 cycles using Phusion Flash High-Fidelity PCR Master Mix (F548L, Thermo Fisher Scientific, Waltham, MA, USA), then 10% (5 µL) of the reaction was used to prepare a 15 µL PCR mix containing SyBR Green in order to estimate the optimal number of amplification cycles (enough to reach between 25% and 33% of maximum fluorescence). Remaining 45 µL were run for the calculated number of cycles. Amplified DNA was purified using Agencourt AMPure XP beads (Beckman Coulter, Brea, CA), according to manufacturer's instructions. Tagmented fragments were isolated and PCR-amplified for 6–9 cycles as described previously[66]. ATAC-seq libraries were sequenced using 42 paired-end sequencing cycles on an Illumina NextSeq500 system at the high throughput sequencing core facility of Institute for Integrative Biology of the Cell (CNRS, France, Centre de Recherche de Gif – http://www.i2bc.paris-saclay.fr/). Reads were demultiplexed using bcl2fastq2-2.18.12, and 31 to 95 million reads (fragments) were obtained per sample. Adaptor sequences were trimmed using CutAdapt v1.15. Raw reads from healthy coronary artery samples were retrieved from sequence read archive (SRR2378591, SRR2378592, SRR2378593).

Analyses were performed on the Galaxy webserver[68]. Reads were mapped on GRCh38 (hg38) genome using Bowtie2 v2.3.4.3 with default settings, except reads could be paired at up to 2 kb distance. Aligned reads were filtered using BAM filter v0.5.9, keeping only mapped, properly paired reads, and removing secondary alignment and PCR duplicate reads as well as blacklisted regions[69]. ATAC peaks were called using MACS2 callpeak v2.1.1.20160309.6 with default settings. Binary read density files (bigwig) were created using bamCoverage v3.3.0.0.0, normalized on hg38 genome.

To perform sample correlation and principal component analyses, a common list of enriched regions was generated using bedtools multiple intersect (Galaxy Version 2.29.0) in "cluster" mode, and average read coverage on these regions was computed using deepTools multiBamsummary (Galaxy Version 3.3.2.0.0). deepTools plotCoverage and plotPCA functions were used to calculate Spearman correlation between samples and Principal Component Analysis, respectively. Global peak annotation was performed using ChIPSeeker v1.22.0[70]. We used Diffbind (Galaxy Version 2.10.0) to detect differentially accessible regions between Artery and SMC and Artery and EC samples[71]. Peaks enriched in arteries in both conditions were taken as artery-specific peaks ($N = 5251$). clusterProfiler v3.14.0 was used to annotate genes at proximity (≤10 kb, 1425 genes) of ATAC-Seq peaks and identify enriched gene ontology terms[72]. Identified GOBP terms were clustered using REVIGO webserver (http://revigo.irb.hr/), with "medium" settings[73]. We used DAVID webserver (https://david.ncifcrf.gov/) to perform functional annotation clustering with default parameters except GO FAT categories were added to the analysis.

**Annotation with epigenomic data**. We computed the overlap of variant with open chromatin regions (narrowpeak from MACS2 output + 100 bp on each side) and histone-ChIp peaks using bedtools (v2.29.0) annotate function. The 95% credible set of causal variants was retrieved from LocusZoom (http://locuszoom.org/). Full list of peak files used is available in Supplementary Table S14. Analysis of SNP enrichment among ATAC-Seq peaks was performed using GREGOR[19]. The lead SNPs from loci

associated with $P$ value $< 10^{-4}$ were used as reference for FMD-associated SNPs (Supplementary Data 1). We included in the analysis SNPs in LD with lead SNPs ($r^2 \geq 0.7$ in the European subset of the 1000 Genomes phase 3 reference panel). We used Integrated Genome Browser (IGB, v9.1.4) to visualize read density profiles and peak positions in the context of human genome[74].

**Overlap between FMD loci and other traits and diseases**. We queried the GWAS catalogue database[75], UK Biobank GWAS summary statistics made publicly available by the Neale lab at the Broad Institute (http://www.nealelab.is/uk-biobank) and GWAS meta-analyses on blood pressure[21], spontaneous coronary artery dissection[24], cervical artery dissection[23], migraine[22], CAD/MI[26], IA/uIA/SAH[76] and Stroke[77] with FMD lead SNPs and LD proxies ($r^2 \geq 0.5$ in the European panel of the 1000 Genomes phase 3). We reported vascular phenotypes with at least one variant with genome-wide ($P < 5 \times 10^{-8}$) or suggestive ($P < 10^{-5}$) association. Colocalization plots were generated using locuscompareR package[62]. Comparative TWAS in FMD and other traits was performed using HapMap filtered summary statistics (see below).

**Heritability and genetic correlation analyses**. We used LD score regression to assess the SNP-based heritability (h2) of FMD and to estimate genetic correlation between FMD and other diseases and traits[12]. Linkage disequilibrium score regression regresses SNP GWAS $\chi^2$ statistics for FMD (to infer SNP-based heritability) or $\chi^2$ statistics cross products for 2 traits (to infer Genetic correlation) on LD scores (the sum of a SNP pairwise squared correlation with other SNPs in a 1 cM window). Summary statistics were acquired from the respective consortia and are detailed in Supplementary Table S15. For each trait, we filtered the summary statistics to the subset of HapMap 3 SNPs to decrease the potential for bias due to poor imputation quality. Correlation analyses were restricted to summary statistics from European ancestry meta-analyses. We used the European LD score files calculated from the 1000G reference panel and provided by the developers. A $P < 1.6 \times 10^{-3}$, corresponding to adjustment for 31 independent phenotypes was considered significant. All analyses were performed with the ldsc package (v1.0.1, https://github.com/bulik/ldsc/). We conditioned FMD association on systolic blood pressure genetic association using multi-trait-based conditional and joint analysis (mtCOJO) tool from GCTA pipeline[78]. The resulting summary statistics were then used to calculate genetic correlation between FMD, conditioned on systolic blood pressure, and the previous traits.

**Reporting summary**. Further information on research design is available in the Nature Research Reporting Summary linked to this article.

## Data availability

The GWAS data generated in this study have been deposited in the GWAS catalogue database under accession code GCST90026612. Summary statistics data used for genetic correlation were acquired from the respective consortia and are detailed in Supplementary Table S15. The ATAC-Seq sequencing data generated in this study are publicly available on the Sequence Read Archive under accession code PRJNA690001. Other genomic datasets used in these studies were retrieved from public databases and a full list of files with accession is available in Supplementary Table S14. Fibroblast eQTL dataset generated was looked up for identified genes only and cannot be made fully available to protect patient's privacy. Data accessibility will be provided upon reasonable request to DEFINE-FMD leaders.

## Code availability

Publicly available softwares and packages were used throughout this study according to developers instruction. No custom algorithms were generated for the study. A complete list of softwares and packages is provided in the reporting summary.

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

## Acknowledgements

We thank all patients who participated in these studies. We thank Dr. Antoine Chédid for collecting and managing clinical data of patients in ARCADIA protocol. We thank Patrick Bruneval for his scientific input and exchanges about arterial pathology in FMD. This study contributes to the IdEx Université de Paris ANR-18-IDEX-0001. This work has benefited from the facilities and expertise of the high throughput sequencing core facility of I2BC (Centre de Recherche de Gif – http://www.i2bc.paris-saclay.fr/). ARCADIA-Pol investigators thank Ewa Rudolf, Elzbieta Pazio and Małgorzata Lewandowska, who were responsible for all administrative work. The Steering Committee of the WOBASZ study expresses special thanks for participating in the implementation of the study to: all their co-workers from research teams at six academic centres, to nurses, doctors, and analysts from field research centres located in 16 voivodeships. We acknowledge the Spanish National Cancer Research Centre (CNIO), in the Human Genotyping lab, a member of CeGen where genotyping was performed for part of the cohorts studied. We thank the participants of the study and the Fibromuscular Dysplasia Society of America for facilitating the enrolment of subjects at their annual meetings. We thank the Frankel Cardiovascular Center and M-BRISC programme for their support and the UM Advanced Genomics Core where genotyping of UM-MGI/CCF samples was performed. The authors acknowledge the University of Michigan Precision Health Initiative and Medical School Central Biorepository for providing biospecimen storage, management, processing and distribution services and the Center for Statistical Genetics in the Department of Biostatistics at the School of Public Health for genotype data management in support of this research. This study was supported by the European Research Council grant (ERC-Stg-ROSALIND-716628) to NB-N and National Institute of Health grant (R01HL139672) to S.K.G. The ARCADIA study was sponsored by the Assistance Publique-Hôpitaux de Paris and funded by a grant from the French Ministry of Health (Programme Hospitalier de Recherche Clinique 2009, AOM 08192) and the Fondation de Recherche sur l'Hypertension Artérielle. Genotyping of French study was supported by the French research agency (ANR-13-JSV1-0002) to N.B.-N. The genotyping of controls from the Three-City Study (3C) was supported by the non-profit organization Fondation Alzheimer (Paris, France) to P.A. ARCADIA-Pol study was supported by the grant no. 2.40/III/19 of Institute of Cardiology, Poland. The WOBASZ II Project was financed from the resources at the disposal of the Polish Minister of Health within the framework of the "National Program of Equalization and Accessibility to Cardiovascular Disease Prevention and Treatment for 2010-2012". M.V. benefited from

Fonds de la Recherche Scientifique - FNRS Grant T.0247.19, Belgium. The Spanish National Cancer Research Centre (CNIO), in the Human Genotyping lab, a member of CeGen Biomolecular resources platform (PRB3), is supported by grant PT17 /0019, of the PE I + D + i 2013-2016, funded by *Instituto de Salud Carlos III* and a European regional development fund (ERDF). DEFINE-FMD is supported by NIH grant 1R01HL148167-01A1 to J.C.K. BAS is supported by the Mayo Clinic Clinician-Investigator Training Program. I.J.K. is additionally supported by NIH grant K24HL137010. The UM-MGI/CCF study is supported by NHLBI/NIH (R01HL139672, R01 HL122684), the University of Michigan Taubman Institute, and Frankel Cardio-vascular Center. S.K.G. is supported by R01HL139672, R01HL122684, and R01HL086694. The Michigan Genomics Initiative (MGI) was supported by the University of Michigan Precision Health Initiative. The Cleveland Clinic Biorepository was supported by CTSA 1UL1RR024989. The Cleveland Clinic FMD Biorepository has been supported in part by the National Institutes of Health, National Center for Research Resources, CTSA 1UL1RR024989, Cleveland, Ohio. Y.R. received funding from the European Research Council (ERC) under the European Union's Horizon 2020 research and innovation program (grant agreement No. 852173). Intracranial aneurysm working group acknowledges the support from the Netherlands Cardiovascular Research Initiative: an initiative with support of the Dutch Heart Foundation, CVON2015-08 ERASE.

## Author contributions

Writing and editing the manuscript: A.G., T.-E.B., S.K.G., N.B-N. Study design/conception: A.G., T.-E.B., S.K.G., N.B-N. Genotyping experiments: J.-F.D., D.D., K.L.H., M.V. Sample /phenotype contribution: L.A., C.A.B., C.M.B., D.M.C., H.G., S.K.G., P.D.L., N.F.-M., D.K.-D., J.Z.L., A.L., M.P., A.P., W.S., J.C.S., M.Z., X.Z., S.Z., FEIRI Consortium, P.A., M.L.d.B., S.D., P.D., W.D., H.L.G., J.W.O., J.P., E.R.R., E.W.-C., A.J., I.J.K., M.A., X.J., A.P., J.C.K. GWAS analyses: M.-L.Y., T.-E.B., M.B., O.D., S.R.K., L.M., B.A.S., S.S., I.S.-S., M.Y., X.Z., J.Z.L. Gene-based and LD score analyses: T.-E.B. Functional annotation experiments: A.G., S.K., L.L. TWAS, in silico functional annotations: A.G. eQTL colocalization analyses: M.-L.Y. Data for IA/SAH genetic correlation: M.B., ISGC intracranial working group, Y.R. Data for stroke genetic correlation: MEGASTROKE. eQTL data and analysis in FMD and control fibroblasts: L.M., V.d'E., J.C.K.

## Competing interests

H.L.G., S.K.G., J.O., and J.C.S. are non-compensated members of the Medical Advisory Board of the FMD Society of America (FMDSA). S.K.G. is a non-compensated member of the Scientific Advisory Board of SCAD Alliance. Both are non-profit organizations.

## Ethics

All studies involved individual written informed consent from all participants and received approval from respective local ethics committee. The ARCADIA study was approved by the « Comité de Protection des Personnes » CPP Ile-de-France II- ID RCB: 2009-A00288-49. The 3 cities protocol was approved by comité consultatif de protection des personnes dans la recherche biomédicale Bicêtre Hôpital Bicêtre n°99-28 CCPPRB approved 10/06/99, 11/03/2003 and 17/03/2006. ARCADIA-Pol study was approved by Local Ethics Committee, Institute of Cardiology, IK-NPIA-0021017/1482/17. The WOBASZ II Project was accepted by the Field Bioethics Committee of the Institute of Cardiology in Warsaw (IK-NP-0021-69/1344/12). All centres included in FEIRI received approval from the respective local/ national ethics committees. ASKLEPIOS study was approved by the ethical committee of the Ghent University Hospital, Belgium. The DEFINE-FMD case control study is Mount Sinai Health System Study ID: HSM# 13-00575/GCO# 13-1118. The Mayo Clinic case control study was approved by Mayo Clinic IRB #08-008355. The UM/CCF case control study was approved by University of Michigan IRB #HUM00044507, #HUM00112101 and Cleveland Clinic IRB approval #10-318.

## Additional information

[1]PARCC, INSERM, Université de Paris, F-750015 Paris, France. [2]Division of Cardiovascular Medicine, Department of Internal Medicine, University of Michigan Medical School, Ann Arbor, MI, USA. [3]Department of Human Genetics, University of Michigan Medical School, Ann Arbor, MI, USA. [4]Department of Neurology and Neurosurgery, University Medical Center Utrecht Brain Center, Utrecht University, Utrecht, The Netherlands. [5]Department of Cardiovascular Medicine, Mayo Clinic, Rochester, MN 55902, USA. [6]Icahn Institute for Genomics and Multiscale Biology, Icahn School of Medicine at Mount Sinai, New York, NY, USA. [7]Centre National de Recherche en Génomique Humaine, Institut de Génomique, CEA and Fondation Jean Dausset-CEPH, Evry, France. [8]Hypertension Unit, Assistance Publique-Hôpitaux de Paris, Hôpital Européen Georges Pompidou, F-75015 Paris, France. [9]Department of Anesthesiology, Michigan Medicine, University of Michigan, Ann Arbor, MI, USA. [10]Vascular Surgery Section, Department of Surgery, Michigan Medicine, University of Michigan, Ann Arbor, MI 48109, USA. [11]Cardiovascular Institute, Icahn School of Medicine at Mount Sinai, New York, NY, USA. [12]Department of Internal Medicine, Division of General Internal Medicine, Section Vascular Medicine, Maastricht University Medical Centre, Maastricht University, Maastricht, the Netherlands. [13]CARIM School for Cardiovascular Diseases, Maastricht University Medical Centre, Maastricht University, Maastricht, the Netherlands. [14]Heart and Vascular Institute, Cleveland Clinic, Cleveland, OH 44195, USA. [15]Zena and Michael A. Wiener Cardiovascular Institute and Marie-Josée and Henry R, Kravis Center for Cardiovascular Health Icahn School of Medicine at Mount Sinai, New York, NY, USA. [16]Division of Cardiology, Cliniques Universitaires Saint-Luc, Université Catholique de Louvain, 1200 Brussels, Belgium. [17]Division of Internal Medicine and Hypertension Unit, Department of Medical Sciences, University of Turin, Turin, Italy. [18]Department of Hypertension, National Institute of Cardiology, Warsaw, Poland. [19]Department of Demography, University of Lodz, Lodz, Poland. [20]Department of Biostatistics and Center for Statistical Genetics, University of Michigan School of Public Health, Ann Arbor, MI, USA. [21]Univ. Lille, Inserm, CHU Lille, Institut Pasteur de Lille, U1167 - RID-AGE - Labex DISTALZ - Risk factors and molecular determinants of aging-related disease, F-59000 Lille, France. [22]Department of Cardiovascular Diseases, Ghent University and Ghent University Hospital, Ghent, Belgium. [23]Department of Neurology, Bordeaux University Hospital, Inserm U1219, Bordeaux, France. [24]Department of Epidemiology, Cardiovascular Disease Prevention, and Health Promotion, National Institute of Cardiology, Warsaw, Poland. [25]Human Molecular Genetics, de Duve Institute, Université Catholique de Louvain, 1200 Brussels, Belgium. [26]Gonda Vascular Center, Mayo Clinic, Rochester, MN 55902, USA. [27]Université de Paris, Inserm, Centre d'Investigation Clinique 1418, F-75006 Paris, France. [28]Department of Genetics, Assistance-Publiques-Hôpitaux de Paris, Hôpital Européen Georges Pompidou, F-75015 Paris, France. [29]Pole of Cardiovascular Research, Institut de

Recherche Expérimentale et Clinique, Université Catholique de Louvain, 1200 Brussels, Belgium. [30]Victor Chang Cardiac Research Institute, Darlinghurst, NSW, Australia. [31]St. Vincent's Clinical School, University of New South Wales, Sydney, NSW, Australia. [32]These authors contributed equally: Adrien Georges, Min-Lee Yang, Takiy-Eddine Berrandou. [33]These authors jointly supervised this work: Santhi K. Ganesh, Nabila Bouatia-Naji. *Lists of authors and their affiliations appear at the end of the paper. ✉email: sganesh@med.umich.edu; nabila.bouatia-naji@inserm.fr

## FEIRI investigators

Peter de Leeuw[12,13], Marco Pappaccogli[16,17], Miikka Vikkula [25] & Alexandre Persu[16,29]

## International Stroke Genetics Consortium (ISGC) Intracranial Aneurysm Working Group

Mark K. Bakker [4] & Ynte M. Ruigrok [5]

## MEGASTROKE

Philippe Amouyel [21] & Stéphanie Debette [23]

## ARCADIA Investigators

Alexandre Persu[16,29], Michel Azizi[8,27] & Xavier Jeunemaitre [1,28]

A full list of members and their affiliations appears in the Supplementary Information.

