## [Peer Review File · Nature Communications]

Genetic investigation of fibromuscular dysplasia identifies risk loci and shared genetics with common cardiovascular diseasesReviewers' Comments:

Reviewer #1:

Remarks to the Author:

Georges and colleagues present a GWAS meta-analysis of fibromusclar dysplasia (FMD) in ~9000 individuals across six cohorts, of which there were 1962 FMD cases. While there are familial inheritance patterns in FMD, there is likely polygenic architecture that needs further characterization. 5 loci were identified (4 through GWAS and one through TWAS). I have the following suggestions and requests for clarification from the authors:

Major:

My first major question is about the characterization of FMD in the studies provided. Were these limited renal FMD patients or were cerebrovascular pts included as well? Was it uniform across studies? In FIERI, those with diagnosis based solely on duplex ultrasonography – but in other studies that does not appear to be case. The diagnostic criteria employed in each study is unclear. Are cases only limited to those who presented with symptoms? The cohorts selected appear to have been enriched for vascular disease patients except for one– given this, how were controls selected? For example, I see controls were participants 'without known FMD'. Is this from self-report or did all controls receive non-invasive imaging for example? This may have to be stated as a limitation. I think clearer presentation of these considerations for case and control would be well suited with a clear table outlining the criteria for each study included in your study characteristics table.

Given the small size of the study already, I was wondering why the investigators didn't run this as a mixed model instead of excluding those who are related?

Can you provide the inflation factor before and after GC correction as well as the LDSR intercept?

It is unclear to me, but was the X chromosome included in this analysis or were those variants excluded? Seems it would be relevant in a condition with differential sex predisposition.

- "Given that the exact prevalence of FMD is unknown in the general population, heritability on a liability scale was estimated between 0.24 ± 0.07 and 0.45 ± 0.13 for a population prevalence ranging from 0.01 to 0.10, respectively." These prevalence numbers seem rather high for FMD in the population. Could you please justify this choice?

- The authors identified loci associated with blood pressure. Did you see if BP loci are associated with FMD? Perhaps existing summary data can be used for two sample Mendelian randomization analyses where you could answer the following two questions: 1) what does FMD cause? 2) what causes FMD?

Minor:

Line 123 appears to have an incomplete sentence: Eighty to 90% of FMD patient present the multifocal phenotype, characterized by a succession of multiple stenosis and aneurysms visible in the an Diagnosis is often made incidentally on imaging..."

In line 163, your woman only GWAS has 1715 cases and 5724 controls, despite there being only 274 men in the study (and your whole study have 1962 cases and 7100 controls). How did the number shrink that much if there were only 274 men? Was this a typo?

Reviewer #2:

Remarks to the Author:

This study identifies novel loci for FMD, through the largest (albeit small) GWAS meta-analysis

available today and provides novel insights into the biological mechanisms of the condition and into the relationship of FMD with other CVD. In general, the manuscript is well written and statistical analyses seem valid. My comments on the study have mostly to do with the presentation of the data. I recommend a more focused approach of presenting the data, including the focus on one main FMD for discovery (remove analyses of other FMD phenotypes, or explore only as secondary analyses in Supplement). Various types of tests are performed in this study with several suggestive reports of various genes, sometimes leaving the reader confused about how to interpret. Move most of the results that do not directly support the main message of the paper to Supplement. Specific comments/questions on the different parts of the study:

GWAS:

GWAS significance threshold for association with FMD of $5E-8$ is applied. Yet, 3 slight variations of the phenotype (FMD all, FMD multifocal, and FMD women) are tested. Significance should be viewed with this in mind. Given that the total number of cases is small to begin with, separate tests for subgroups that represent the majority of cases; multifocal (91% of cases) and women (87% of cases) should not be applied for the purpose of identifying novel loci, i.e. loci that are not significant in the main FMD phenotype. To the reviewer this appears only to "squeeze out" more loci through multiple testing, which is not convincing.

What is the rationale behind the different cut-offs for removing SNPs with low imputation quality (< 0.8 for French and UM studies and < 0.3 for the others studies)? Why not use the 0.8 for all studies? What is the info for the main reported GWAS significant variants per study?

Gene-based analysis:

The authors claim to have identified four genes through the gene-based analysis, from two different loci to be associated with FMD ($P_{Bonf.Adj} < 0.05$) in at least one of the tested samples (all, multifocals, women) (LIMA1, $P_{Women} = 3.0 \times 10^{-7}$, ATF1, $P_{Multifocal} = 7.3 \times 10^{-7}$, and GPD1, $P_{Multifocal} = 7.3 \times 10^{-7}$) and STAT6 ($P_{Multifocal} = 2.4 \times 10^{-6}$), near LRP1. For me this claim is confusing – in particular when the authors conclude in discussion "Key findings from our study include: i) evidence for polygenic feature of FMD and three new risk loci and/or genes using single variant and gene-based GWAS, and TWAS in arteries;" What are the 3 novel risk loci and/or genes are the authors referring to as the key findings ?

For the same reasons as for the GWAS, and to avoid multiple testing issues, use only one main FMD phenotype for discovery (the same phenotype as used for GWAS discovery).

Explain in more detail what the gene-based analysis does. How do the authors interpret the results. Are the authors claiming that these genes are novel FMD genes? Should "significant" P_{adj} results for SNPs in a particular gene be interpreted as the gene is a disease-causing gene and non-significant to be interpreted as the gene is not causally affecting the disease? Are the lead SNPs at each locus included in the gene-based analysis? Help the reader to understand why PHACTR1 gene is not significant after Bonferroni correction? Is the suggestive association of STAT6 SNPs independent from the lead SNP at the locus (rs11172113 – that points to the LRP1 gene through cis eQTL) ? Can SNPs positioned in one gene not be affecting other locus genes, i.e. not necessarily affecting the gene it is positioned in ? If so, what additional information does this analysis provide for the current data?

TWAS:

The authors report significant association between genetically predicted expression of PHACTR1 ($P = 1.1 \times 10^{-11}$, tibial and aorta), LRP1 ($P = 2.7 \times 10^{-10}$, tibial) and ATP2B1 ($P = 3.7 \times 10^{-6}$, tibial). For PHACTR1 and ATP2B1 the TWAS significance is considerably less than for the lead SNPs at each locus (which are the top cis eQTL for the respective genes). Does this mean that other eQTL SNPs (independent from the lead SNPs) do not associate with disease, hence the lower significance? How do the authors interpret this?

Consider shortening the section on gene-based analysis and TWAS and move most of the results to Supplement. Use one main FMD phenotype as the main analysis to avoid multiple testing issues.

FMD and blood pressure:

Since all the FMD loci are previously reported to associate with blood pressure traits it is worth investigating how many of established GWAS blood pressure loci ($N \sim 1000$) associate with FMD (at P adjusted for the number of tests)? Provide a plot of the effects of the blood pressure SNPs on blood

pressure vs their effect on FMD. Is there a clear correlation between the effect sizes on blood pressure and on FMD ? Even though FMD has been viewed as a cause of (contributing to) hypertension, is the data consistent with hypertension being a causal factor of FMD, or is the data more consistent with shared genetic factors contributing to both ? Does that mean that FMD is not a causal factor for hypertension ? Discuss.

RESPONSES TO REVIEWER COMMENTS

Reviewer #1 (Remarks to the Author):

Georges and colleagues present a GWAS meta-analysis of fibromuscular dysplasia (FMD) in ~9000 individuals across six cohorts, of which there were 1962 FMD cases. While there are familial inheritance patterns in FMD, there is likely polygenic architecture that needs further characterization. 5 loci were identified (4 through GWAS and one through TWAS). I have the following suggestions and requests for clarification from the authors:

Major:

My first major question is about the characterization of FMD in the studies provided. Were these limited renal FMD patients or were cerebrovascular pts included as well? Was it uniform across studies?

The majority of patients presented with either reno-vascular or cerebrovascular FMD, but other vascular beds were also involved. This was the case in all cohorts studied. To clarify this point, we have added a new column to the Supp. Table S1.

In FIERI, those with diagnosis based solely on duplex ultrasonography – but in other studies that does not appear to be case. The diagnostic criteria employed in each study is unclear. Are cases only limited to those who presented with symptoms?

All studies utilized angiographic methods (invasive with DSA or non-invasive, such as by CTA or MRA) to confirm a diagnosis of FMD. No individuals were included with FMD diagnosis based on duplex ultrasound findings only. To clarify this in the Methods, we extracted the imaging methods used from cited references for cohorts where this information was not detailed. We now explicitly indicate the imaging method in all cohorts' individuals descriptive (Supp Material, Methods section).

The cohorts selected appear to have been enriched for vascular disease patients except for one—given this, how were controls selected? For example, I see controls were participants ‘without known FMD’ Is this from self-report or did all controls receive non-invasive imaging for example? This may have to be stated as a limitation.

I think clearer presentation of these considerations for case and control would be well suited with a clear table outlining the criteria for each study included in your study characteristics table.

Depending on case control studies, controls were either ascertained from country matched population-based cohorts (3C Study, WOBASZ II and ASKLEPIOS), recruited specifically for the purpose of the study (DEFINE Study) or extracted from local clinical registries linked to electronic health record data (MGI, Mayo Clinic). Imaging to definitively exclude FMD in controls was not performed, as the risks (radiation and/or contrast dye) are not justified. Participants in the control group were not specifically enriched in vascular disease, and individuals with known FMD or other vascular disease (from clinical record) were excluded (MGI, Mayo Clinic, DEFINE-Study); however given currently accepted estimated prevalence of

FMD as approximately 3%, a small proportion of controls may have in fact had subclinical disease. This may result in a dilution of the effect sizes found. We now include this limitation in the revised article. (page 21)

We have also updated our clinical characteristics table accordingly (Supp. Table S1).

Given the small size of the study already, I was wondering why the investigators didn't run this as a mixed model instead of excluding those who are related?

The recruitment of patients was not family-based and the cohorts of cases studied were marginally enriched for related individuals (less than 4%). We anticipated a limited gain in power by keeping those individuals and applied logistic regression in all individual case control studies. We also note that the inclusion of related individuals would have limited subsequent LD score regression analyses.

Can you provide the inflation factor before and after GC correction as well as the LDSR intercept?

This information is now added to Supplementary Table S15 (Quality Control)

Study	lambda GC	meta-analysis after correcting per study for lambda GC		meta-analysis without correcting per study for lambda GC	
		Lambda GC	LDSR intercept	Lambda GC	LDSR intercept
FR	1.02				
Mayo	0.97				
Define	0.95				
Pol	0.96	1.04	1.03	1.05	1.04
UM	1.01				
Feiri	1.03				

It is unclear to me, but was the X chromosome included in this analysis or were those variants excluded? Seems it would be relevant in a condition with differential sex predisposition.

Chromosome X genetic association results were not displayed in our initial submission. We have now provided results from Chr X in the revised manuscript. Our results do not support, at least in the current cohorts, the existence of a risk locus on X chromosome (updated Figure 1a).

- "Given that the exact prevalence of FMD is unknown in the general population, heritability on a liability scale was estimated between 0.24 ± 0.07 and 0.45 ± 0.13 for a population prevalence ranging from 0.01 to 0.10, respectively." These prevalence numbers seem rather high for FMD in the population. Could you please justify this choice?

FMD has long been perceived as a rare disease but in modern registries and data based upon renal donor studies of the renal artery in pre-transplant evaluation, we have more recently learned that FMD is in fact not rare in adults. The reasons for the under-diagnosis of FMD are complex and are understood to relate to vagueness of certain symptoms occurring in young

women without “traditional” cardiovascular risk factors. This is reflected in the average delay to diagnosis > 4 years. As another likely explanation, FMD may be mild and sub-clinical (i.e. is does not cause any symptoms or phenotypic features). Further, FMD is an angiographically diagnosed phenotype, and imaging surveillance required to make a diagnosis has only recently been implemented.

Several studies have examined series of catheter-based or computed tomographic angiography imaging performed on kidney donors prior to surgery and found a prevalence of asymptomatic FMD ranging from 2 to 6% in this a priori healthy population. Shivapour DM et al, *Vasc Med.* 2016, cited as Reference #10 in the manuscript, summarizes these findings, and this article is now more clearly emphasized in the introduction of the revised manuscript. We now write that the currently best-accepted estimate is ~3%, although we acknowledge this as an approximation that may not reflect the prevalence of symptomatic FMD. Nonetheless, we now report the heritability estimate for FMD on a liability scale for a prevalence of 3% to be 0.43 ± 0.14 . (Results page 8 lines 158-159)

- The authors identified loci associated with blood pressure. Did you see if BP loci are associated with FMD?

From the updated summary statistics of SBP released by Evangelou et al, *Nat Genet* 2018, we identified 821 independent loci ($MAF \geq 0.01$, $LD r^2 = 0.01$ and window = 1Mb) associated with SBP at GWAS significance threshold. Among these, 3 loci (PHACTR1, LIMA1 and ATP2B1) were associated with FMD at GWAS significance threshold and 6 at a suggestive significance level ($p < 0.001$). One locus is statistically significantly associated with FMD considering adjustment for multiple testing correction in this lookup analysis (near FHL5 on chromosome 6), and we included this information in the main text page 13 and in Supplementary Table S8.

Perhaps existing summary data can be used for two sample Mendelian randomization analyses where you could answer the following two questions: 1) what does FMD cause? 2) what causes FMD?

We agree that Mendelian randomization (MR) is a powerful association tool to disentangle the putative causal associations between traits using SNPs as instruments. However, we would like to draw the attention of our reviewers to the important limitations of conducting MR in our case: 1) our study is based on a several degrees of magnitude smaller sample size compared to the dataset that we had available to us for SBP to conduct the bi-directional MR we describe below. As a result, the precision in estimation of effects of >800 SBP SNPs is much larger, compared to the precision of estimation of effects for our 4 loci. 2) FMD is a categorical trait while SBP is a continuous trait, which further impacts power. 3) We cannot estimate the directionality of association because FMD status is not available in the UKBB cohort that we used for SBP GWAS summary statistics, and blood pressure is not available in all cases and controls of the FMD GWAS.

We attempted however to explore if bi-directional MR analyses using summary statistics for SBP from the UKBB and the current FMD meta-analysis would help us address the important and interesting questions raised by this Reviewer: what does FMD cause and what causes FMD?

We applied 4 different MR methods: Inverse weighted variance (IVW), MR-Egger, weighted median and weighted mode (each assuming different set of assumptions, see Burgess et al 2013, Bowden et al 2015,2016 and Hartwig et al 2017 for more details) in order to triangulate evidence of causal association existence through replication of results.

1) MR results: SBP → FMD

First, we ran MR on the potential causal path SBP → FMD under the assumption of BP as a risk factor for FMD. Below the plot of SNPs effects on FMD compared to SNPs effects on SBP for the 434 SNPs ($p \leq 5 \times 10^{-8}$ & LD $r^2 < 0.001$ /window: 1Mb) used in this analysis (Reviewer Figure 1, Reviewer Table 1, Reviewer Table 2).

Reviewer Figure 1.

Reviewer Table 1. Two-sample MR result: MR Method

MR method	n_{snp} ($p \leq 5 \times 10^{-8}$ & LD $r^2 < 0.001$)	β	SE	P
Inverse variance weighted	434	0.08	0.01	2.56E-18
MR Egger	434	0.15	0.02	4.98E-11
Weighted median	434	0.08	0.01	6.49E-11
Weighted mode	434	0.08	0.03	3.83E-03

Reviewer Table 2: Two-sample MR result: Heterogeneity test

Method	Q	Q df	Q pval
Inverse variance weighted	602.6182	433	1.00E-07
MR Egger	585.5079	432	1.10E-06

We found a statistically significant association between SBP and FMD using all 4 MR methods. However, we observed significant heterogeneity both for IVW and MR egger methods.

Next, we tested if the SBP →FMD was the right direction to consider in the association between FMD and SBP. We chose not to apply Steiger test (Hemani et al 2017), which ignores the fact that the sample sizes of the two studies are extremely divergent (FMD<10k and SBP: >700k). As an alternative, we compared the sum of betas from SBP GWAS ($\sum_i \beta(SBP)^2_i - SE(SBP)^2_i$) and those from FMD GWAS ($\sum_i \beta(FMD)^2_i - SE(FMD)^2_i$, where "i" runs through all the SNPs which are SNP instruments). As expected given the larger number of SNPs known for SBP, we found a larger sum for SBP (51,00) than for FMD (0.65). This test supports that the directionality of the SBP->FMD is potentially the causal path.

2) MR results: FMD → SBP

In the second step we analysed the alternative path FMD →SBP, considering SBP as a potential consequence of FMD.

Here, we only used 4 lead SNPs from 4 GWAS significant loci that we describe in the article as instruments. The first limitation encountered was validity of FMD →SBP MR analysis using FMD lead SNPs (4 SNPs) as instruments where, 3 out of the 4 SNPs are also associated with SBP. To do this, we checked the effects for each SNP on FMD and SBP by calculating the effects taking into account the sample size according to the following equations FMD effect =

$\frac{1}{\sqrt{N_{FMD}}} \times \frac{\beta_{FMD}}{SE_{FMD}}$ Vs FMD effect = $\frac{1}{\sqrt{N_{SBP}}} \times \frac{\beta_{SBP}}{SE_{SBP}}$. Results in Reviewer table3 show that the effects estimated on FMD are larger for all 4 SNPs than the ones estimated on SBP, which may support a possible causal path from FMD to SBP.

Reviewer Table 3.

SNP	gene	beta.fmd	se.fmd	beta.sbp	se.sbp	fmd_effect	sbp_effect
rs11172113	LRP1	-0.27	0.04	0.05	0.03	-0.07	0.002
rs17249754	ATP2B1	-0.34	0.06	-0.84	0.04	-0.06	-0.02
rs6580732	LIMAI	-0.22	0.04	-0.36	0.03	-0.06	-0.01
rs9349379	PHACTR1	-0.33	0.04	-0.27	0.03	-0.08	-0.01

When we applied two samples MR, three out of 4 SNPs (or instruments) showed evidence for a causal effect from FMD toward higher SBP (Reviewer Table 4).

Reviewer Table 4. Single SNP MR

Gene	SNP	exposure	outcome	β (FMD \rightarrow SBP)	se	p
LRP1	rs11172113	FMD	SBP	-0.18	0.11	0.12
ATP2B1	rs17249754	FMD	SBP	2.46	0.12	1.6E-97
LIMAI	rs6580732	FMD	SBP	1.66	0.14	1.6E-31
PHACTR1	rs9349379	FMD	SBP	0.81	0.09	1.4E-17

When using all 4 instruments together (Reviewer Figure 2 and Reviewer Table 5), causal significance was not observed with most MR methods we applied.

Reviewer Figure 2.

Reviewer Table 5. Results from two sample MR

MR method	nsnp ($p \leq 5 \times 10^{-8}$ & LD $r^2 < 0.001$)	β	SE	P
Inverse variance weighted (IVW)	4	1.08	0.55	0.05
MR Egger	4	1.99	3.79	0.65
Weighted median	4	0.33	0.10	0.001
Weighted mode	4	0.19	0.11	0.19

In addition, we observed important heterogeneity in our setting using IVW and MR-Egger (Reviewer Table 6).

Reviewer Table 6. Heterogeneity test

Method	Q	Q df	Q pval
Inverse variance weighted	286.4169	3	8.66e-62
MR Egger	278.0722	2	4.14e-61

***In summary:** The causal association SBP → FMD was statistically significant, including loci that are associated both with FMD and SBP, although the effect is rather small. The reverse causal association was inconclusive as we were only able to test 4 FMD instruments, obtained from current knowledge about FMD genetic basis.*

In addition to this limitation, the Bi-directional-MR results also suffer from the scale difference between FMD (binary) and SBP (continuous). Also, as mentioned in the introduction of this analysis, the two GWAS have very different power, and it is certainly easier to detect very tiny SBP → FMD causal effects, and we do lack the needed power to detect equivalent small effects of FMD instruments on SBP. Finally, we note that 3 of our 4 instruments for FMD are well established pleiotropic loci associated with a large number of cardiovascular diseases and traits, which has certainly complicated the application of MR in our case.

In the light of all these caveats, we decided not to report these results in the revised manuscript. We fear that the inconclusive results will be misleading information to our readers and will add complexity to the integration of the numerous results we obtained from this first GWAS of FMD. Further, based upon clinical knowledge of FMD, the relationship of FMD and its manifestations to blood pressure is complex, and this is matched by the results of the MR analyses, since hypertension is not only a result of renovascular FMD, but it is also a risk factor for aneurysms and dissections, which occur in 41% of individuals with FMD and are a major presenting finding leading to FMD diagnosis. We provided a discussion about the limitations of the interpretation of the detection of several SBP loci, and a significant genetic correlation between SBP and FMD (Page 19). We also mentioned in the discussion the limitations about the application of MR analysis.

Minor:

Line 123 appears to have an incomplete sentence: Eighty to 90% of FMD patient present the multifocal phenotype, characterized by a succession of multiple stenosis and aneurysms visible in the an Diagnosis is often made incidentally on imaging...”

Thank you, this sentence was corrected.

In line 163, your woman only GWAS has 1715 cases and 5724 controls, despite there being only 274 men in the study (and your whole study have 1962 cases and 7100 controls). How did the number shrink that much if there were only 274 men? Was this a typo?

This number referred to men among FMD cases only, and not in the whole study. This was clarified and numbers were updated (page 8 line 178-180) to match with the current analysis reported.

Reviewer #2 (Remarks to the Author):

This study identifies novel loci for FMD, through the largest (albeit small) GWAS meta-analysis available today and provides novel insights into the biological mechanisms of the condition and into the relationship of FMD with other CVD. In general, the manuscript is well written and statistical analyses seem valid. My comments on the study have mostly to do with the presentation of the data. I recommend a more focused approach of presenting the data, including the focus on one main FMD for discovery (remove analyses of other FMD phenotypes, or explore only as secondary analyses in Supplement). Various types of tests are performed in this study with several suggestive reports of various genes, sometimes leaving the reader confused about how to interpret. Move most of the results that do not directly support the main message of the paper to Supplement.

Specific comments/questions on the different parts of the study:

GWAS:

GWAS significance threshold for association with FMD of $5E-8$ is applied. Yet, 3 slight variations of the phenotype (FMD all, FMD multifocal, and FMD women) are tested. Significance should be viewed with this in mind. Given that the total number of cases is small to begin with, separate tests for subgroups that represent the majority of cases; multifocal (91% of cases) and women (87% of cases) should not be applied for the purpose of identifying novel loci, i.e. loci that are not significant in the main FMD phenotype. To the reviewer this appears only to “squeeze out” more loci through multiple testing, which is not convincing.

We find this comment highly relevant. To avoid the multiple testing issues mentioned, and to simplify the analyses and main results, we have decided to report discovery GWAS, gene-based and TWAS results only for the multifocal FMD phenotype that vascular clinicians who participate in this study agree to be the most relevant and common form of stenosis reported in FMD patients. Results were updated appropriately in all sections of the revised manuscript. We now only report the results of the top SNPs on associated loci in women to appreciate the potential differences in effects sizes estimates in this main group of patients.

What is the rationale behind the different cut-offs for removing SNPs with low imputation quality (< 0.8 for French and UM studies and < 0.3 for the others studies)? Why not use the 0.8 for all studies? What is the info for the main reported GWAS significant variants per study?

We apologize for this confusing situation. The 0.3 criterion for imputation cut-off was initially applied but, in some cohorts, this threshold did not eliminate several loci where imputation results did not match expected ones in the reference panels. To avoid confusion, we now apply the 0.8 criterion for all studies. This led to a ~15% reduction of the number of SNPs tested in the meta-analysis, with no major change in the main findings. However, some of the top associated variants at the LIM1 locus were filtered out, including the previous lead SNP at this locus. We note that none of the top associated variants in this locus that we annotated as most likely functional were affected by this filter and were all kept in the updated dataset.

Gene-based analysis:

The authors claim to have identified four genes through the gene-based analysis, from two different loci to be associated with FMD (P_{Bonf.Adj}<0.05) in at least one of the tested samples (all, multifocals, women) (LIMA1, P_{Women}=3.0×10⁻⁷, ATF1, P_{Multifocal}=7.3×10⁻⁷, and GPD1, P_{Multifocal}=7.3×10⁻⁷) and STAT6 (P_{Multifocal}=2.4×10⁻⁶), near LRP1. For me this claim is confusing – in particular when the authors conclude in discussion “Key findings from our study include: i) evidence for polygenic feature of FMD and three new risk loci and/or genes using single variant and gene-based GWAS, and TWAS in arteries;” What are the 3 novel risk loci and/or genes are the authors referring to as the key findings?

For the same reasons as for the GWAS, and to avoid multiple testing issues, use only one main FMD phenotype for discovery (the same phenotype as used for GWAS discovery).

To prevent the rightly highlighted confusion in reporting our results, we have now limited the analyses to multifocal FMD. The revised manuscript was modified to gain clarity in summarizing the main findings for the gene-based analysis as well. The updated analysis, excluding non-multifocal patients and SNPs below the R_{sq} of 0.80 in all studies resulted in a slightly different set of genes to associate with FMD. Mainly, STAT6 is no longer among this list, while we now report ATP2B1 and one additional gene at the LIMA1 locus (CERS5) (See Supp Table S4). The corresponding Results sections were updated accordingly in the revised manuscript.

Explain in more detail what the gene-based analysis does. How do the authors interpret the results. Are the authors claiming that these genes are novel FMD genes? Should “significant” P_{adj} results for SNPs in a particular gene be interpreted as the gene is a disease-causing gene and non-significant to be interpreted as the gene is not causally affecting the disease?

As reported by the developers of MAGMA algorithm, gene-based analysis is a statistical method for analysing multiple genetic markers simultaneously to determine their joint effect. This method can be used when the effects of individual markers are too weak to stand alone, which is a common problem when studying polygenic traits where individual small effects are the rule. In a gene-based analysis, genetic marker data is aggregated to the level of whole genes, testing the joint association of all markers in the gene with the phenotype. Clearly, the idea is to try and detect signals that would be ignored by the classic SNP-based association, where we lack power to detect them in the individual SNPs-based GWAS. Conversely, as for SNPs association, the identification of positively-associated genes indicates the presence of signals physically mapping to these genes, but does not provide direct evidence of these associated genes to be the biological target genes, although the current accepted consensus in the field is that the closest genes are the most likely target genes. As an example, in the case of associated genes at the LIMA1 locus, the association of all four genes we report is due to the same signal, involving the same set of associated SNPs. At this stage, we are not able to declare any of these 4 genes as the putative causal gene based only on the MAGMA results.

To clarify this information, we have modified the Results and Discussion sections to better emphasize that the main results of this analysis are that we did not identify additional loci involving less significant individual SNPs mapping to specific genes (Results section, Page 10, L200). We have also removed the sentence where we described the gene-based analysis as a source of discovery of FMD associated genes from the Discussion (Page 17, L387-L392).

Are the lead SNPs at each locus included in the gene-based analysis? Help the reader to understand why PHACTR1 gene is not significant after Bonferroni correction?

We confirm that lead SNPs at each locus are included if they map to genes (located physically between, transcription start site and transcription stop site (the default option in MAGMA)). Thus, in the case of PHACTR1 and LRP1 lead SNPs were included in the gene-based analyses we report. The lack of association for these two genes is due to the fact that these association loci both involve a highly associated SNP, with high posterior probability to be the causal variant (>99% in each case). Given that the gene-based algorithm considers multiple association signals to calculate gene association, PHACTR1 and LRP1 were not associated at the gene-level. As mentioned in the previous paragraph, the aim of the gene-based analysis was to detect additional diffuse signals around genes as an association unit. Individual SNPs association is to be considered the most relevant results for association with FMD.

Is the suggestive association of STAT6 SNPs independent from the lead SNP at the locus (rs11172113 – that points to the LRP1 gene through cis eQTL)? Can SNPs positioned in one gene not be affecting other locus genes, i.e. not necessarily affecting the gene it is positioned in? If so, what additional information does this analysis provide for the current data?

The previously statistically significant association of STAT6 was probably caused at least in part by variants in high LD with rs11172113, which is intronic to LRP1. We note that this association signal was lost in the updated analysis after filtering out variants with $r^2 < 0.8$. The currently analysed set of SNPs do not support a gene association for STAT6. We have updated the Results and Discussion accordingly.

TWAS:

The authors report significant association between genetically predicted expression of PHACTR1 ($P=1.1 \times 10^{-11}$, tibial and aorta), LRP1 ($P=2.7 \times 10^{-10}$, tibial) and ATP2B1 ($P=3.7 \times 10^{-6}$, tibial). For PHACTR1 and ATP2B1 the TWAS significance is considerably less than for the lead SNPs at each locus (which are the top cis eQTL for the respective genes). Does this mean that other eQTL SNPs (independent from the lead SNPs) do not associate with disease, hence the lower significance? How do the authors interpret this?

The TWAS association considers all SNPs that are part of the locus and are correlated to the expression of the gene to simulate, with different models the expression changes occurring in the disease analysed.

One possibility to explain the reduced significance for PHACTR1 and ATP2B1 is the potential existence of additional SNPs contributing in the opposite direction, compared to the main signal. For example, at the PHACTR1 locus, the risk allele of the lead variant associated to FMD (rs9349379) is associated with higher expression of PHACTR1. We note that another eQTL variant (rs202401) from an independent LD block presented an opposite effect on gene expression ($\beta = -0.02$, $P=0.07$, with the risk allele for FMD associated with lower expression). The TWAS summarizes a global effect of all SNPs, which we believe is a potential explanation of the differences highlighted here. Of note, ATP2B1 is now below significance threshold (TWAS Bonf correction) when we only keep multifocal FMD and SNPs with $Rsq < 0.80$ in all cohorts.

Consider shortening the section on gene-based analysis and TWAS and move most of the results to Supplement. Use one main FMD phenotype as the main analysis to avoid multiple testing issues.

We thank the reviewer for this very useful suggestion. We have now focused the results on multifocal FMD only in the revised manuscript, which provided an important simplification in the report of gene-based and TWAS results in particular and helped gain fluidity in reporting the results overall.

FMD and blood pressure:

Since all the FMD loci are previously reported to associate with blood pressure traits it is worth investigating how many of established GWAS blood pressure loci (N~1000) associate with FMD (at P adjusted for the number of tests)?

We added the results of this lookup for 645 top SNPs from Evangelou et al. (ref 21) that were present in our study in Supplementary Table 8. One new locus on chromosome 6 (close to FHL5) could be considered statistically significant in this lookup, while two more loci could be considered suggestive ($p < 0.001$). We now mention this result in the Results section (p13, L343)

Provide a plot of the effects of the blood pressure SNPs on blood pressure vs their effect on FMD. Is there a clear correlation between the effect sizes on blood pressure and on FMD?

Reviewer Figure 3

There is indeed a positive and statistically significant (correlation=0.41, $P=4 \times 10^{-27}$) correlation between effects on blood pressure and effects on FMD considering only SBP-significant loci. However, we cannot draw any definite conclusions from these comparisons, given the important lack of precision of estimated of effects on FMD due to our smaller samples, compared to UKBB

sample, where effects on SBP were much more accurately estimated. Nonetheless, this comparison shows deviation from the correlation line of many loci.

Even though FMD has been viewed as a cause of (contributing to) hypertension, is the data consistent with hypertension being a causal factor of FMD, or is the data more consistent with shared genetic factors contributing to both? Does that mean that FMD is not a causal factor for hypertension? Discuss.

The relationship between the presence of FMD and high blood pressure is certainly complex. In our experience with FMD, several different scenarios may be observed clinically in affected patients:

1. The presence of a tight renal stenosis due to FMD and its hemodynamic impact is the cause of increased blood pressure and onset of hypertension, which is in this case qualified as renovascular hypertension. Percutaneous angioplasty to relieve the obstruction is a treatment for renovascular hypertension.

2. Given the high prevalence of essential hypertension, the two diseases can randomly coexist in the same patient, without a putative causal relationship between the presence of FMD and arterial hypertension. In this case, FMD lesions are often described as less severe stenosis and percutaneous angioplasty is less effective or may have no effect on high blood pressure. (see Trinquart et al., Hypertension 2010 for more details about effects on angioplasty on FMD). Referral of patients with unexplained hypertension to hypertension centers of excellence with higher awareness of this disease may also facilitate the detection of FMD in this subpopulation compared to the general population.

3. When FMD lesions are initially diagnosed outside the renal bed (e.g in cerebrovascular arteries), most cases also present with renal FMD, as the involvement of multiple arterial beds is common in FMD, although these patients are less likely to have hypertension (around 56%, instead of 88% compared to renal presentation, Plouin Hypertension 2017). Of note, diagnosis in the cerebrovascular bed is often in the context of stroke events, neurologic symptoms or symptoms owing to altered blood flow (e.g. pulsatile tinnitus). High BP being a risk factor for stroke, this adds an additional layer of complexity to interpret the relationship between FMD and high BP.

4. Finally, hypertension is a risk factor for arterial aneurysms and dissections, which occur in 30-40% of individuals with FMD (Gornik et al., reference 4 in the manuscript), and may be a presenting feature of FMD. Thus, the ascertainment of symptomatic individuals may not represent the true prevalence of hypertension among all individuals having FMD.

In addition to these clinical complexities, our attempt to use the current results to conduct bi-directional Mendelian Randomization failed to clarify this relation due to numerous caveats: mainly the unbalanced current knowledge about the genetics of FMD (4 loci) compared to SBP (>800) and several order of magnitude differences between the power in samples used in GWAS for SBP compared to FMD. We have updated the Discussion of the revised manuscript to reflect these complexities and limitations of the current knowledge to drive conclusive remarks about

the etiological relation between FMD and high BP. We discussed this page 19 and added it as a limitation to our study page 21.

Reviewers' Comments:

Reviewer #1:

Remarks to the Author:

No further comments. Thank you

Reviewer #2:

Remarks to the Author:

The reviewer is still concerned that some of the analyses performed are given too much weight in the paper without being significant or very informative. Although the reviewer has not assessed the word-restrictions given by the Journal, the manuscript appears/feels too long. Reading the manuscript gives the impression that the authors perform a variety of analyses, just to tick boxes. Some of these analyses, particularly those that are of borderline significance, where interpretations are difficult to make, are described in too much detail.

Thus the reviewer still recommends a more focused approach of presenting the data, involving a major reconstruction of the manuscript, cutting out or de-emphasizing many of the results.

Specific comments/questions on the different parts of the study:

In the INTRODUCTION, this is stated: "Through the integration of open chromatin maps generated by ATAC-Seq in fibroblasts, smooth muscle and endothelial cells, combined with public resources available in arteries, we prioritized variants and confidently narrow down target genes in associated loci".

While the authors did identify most likely causal variants at some of the loci through this method, they cannot claim to have confidently narrowed down target genes in associated loci through these methods.

The following claim is not valid. On what grounds/results is it made?:

We found that risk genes for FMD are specifically and consistently expressed in smooth muscle cells, fibroblasts, and arterial tissues and are involved in regulatory mechanisms related to actin cytoskeleton and intracellular calcium homeostasis, a mechanism central to vascular contraction. For example, PHACTR1 and LRP1 are widely expressed (including brain, lung, whole blood, skin, adipose; see GTex) not specifically in the mentioned tissues. Additionally, while you found that artery-specific ATAC-Seq peaks are overrepresented in the vicinity of genes involved in contractile fibres and muscle system processes in general, you did not establish through your analyses that FMD genes are involved in these processes.

Gene-based analysis:

If significant, the gene-based analysis only indicates the presence of signals physically mapping to genes, but the analysis does not provide direct evidence of these associated genes to be the biological target genes.

Thus, in RESULTS, listing up genes along with insignificant P-values resulting from the analysis has no meaning. Further, the results would not point to potential causal genes, even if they were significant! To avoid reader distraction, the reviewer suggests to shrink the paragraph in results (or even better to remove altogether) to one sentence saying something like the gene-based analyses using MAGMA did not identify additional association signals (refer to results in supplement).

FMD associated variants are likely to regulate gene expressions of nearby genes in arterial tissues: In this results section, the lead variants at the PHACTR1, LRP1, and ATP2B1 loci are all correlated with the top cis-eQTL for these genes indicating that the FMD signals may be driven through an effect on expression of these genes. However, in the next results section "Transcriptome-wide association study in arteries" association between expression of PHACTR1 and LRP1 with the disease are confirmed, but the authors state that the association of ATP2B1 is not significant. What is the conclusion/interpretation then regarding ATP2B1?? Do the authors doubt that ATP2B1 is the causal gene behind the signal given these results?

In table 2, the authors should present the lead SNP at the locus rather than presenting results for gene-based association. According to comment above, the signals resulting from gene-based analysis are not providing superior evidence (to the lead SNP) in support of the gene being the causal gene.

The following results are not very convincing (borderline significance), in fact they make little sense and only confuse the reviewer. Why would the variant only affect expression in patients but not in non-FMD patients? Consider removing this sentence. After all, expression of PHACTR1 associates with rs9349379 in non-FMD patients in GTex:

Interestingly, a query of transcriptome sequencing data obtained in primary dermal fibroblasts cell lines from 83 FMD patients and 70 matched controls found rs9349379 to correlate with PHACTR1 expression in the FMD group ($P=0.01$), but not the control group (Supplementary Figure S3).

FMD GWAS variants mapping to ATAC-Seq peaks in arterial primary cells and tissues

I find that these results are given too much space (and interpretation) given that they do not seem significant (P -values from 0.07 -0.01 for enrichment of lead variants that are not significant to begin with (restricting on $P<10^{-4}$ and correlated variants with $r^2>0.7$). I strongly recommend to shrink this section considerably (perhaps move to Supplement), or even better, to remove altogether.

FMD potential target genes are expressed in VSMCs and fibroblasts

I think this section should be shortened considerably – the message being that the PHACTR1, LRP1, and other candidate genes are expressed in tissues that are relevant for FMD.

Further, the reviewer finds these results somewhat contradicting the results where the authors claim to have found enrichment (although not highly significant) of lead variants in open chromatin peaks of artery tissues, but NOT in SMC and fibroblasts.

I am further sceptic about whether the potential association of candidate gene expression with sex can be considered significant, and I would certainly not suggest that PHACTR1 and ATF1 had sex-specific expression. These results can be removed without harming the overall story.

Discussion:

Use a more descriptive wording for the following: "ii) through integration of annotation datasets generated inhouse", i.e. in what house was it generated ?

Please revise/reconsider the following claim (see comment above):

iii) FMD risk genes showed consistent and specific expressions in VSMCs, fibroblasts and arterial tissue and were involved in regulatory mechanisms related to actin cytoskeleton and intracellular calcium homeostasis, a mechanism central to vascular contraction.

REVIEWER COMMENTS

Reviewer #1 (Remarks to the Author):

No further comments. Thank you

Reviewer #2 (Remarks to the Author):

The reviewer is still concerned that some of the analyses performed are given too much weight in the paper without being significant or very informative. Although the reviewer has not assessed the word-restrictions given by the Journal, the manuscript appears/feels too long. Reading the manuscript gives the impression that the authors perform a variety of analyses, just to tick boxes. Some of these analyses, particularly those that are of borderline significance, where interpretations are difficult to make, are described in too much detail.

Thus the reviewer still recommends a more focused approach of presenting the data, involving a major reconstruction of the manuscript, cutting out or de-emphasizing many of the results.

Specific comments/questions on the different parts of the study:

In the INTRODUCTION, this is stated: “Through the integration of open chromatin maps generated by ATAC-Seq in fibroblasts, smooth muscle and endothelial cells, combined with public resources available in arteries, we prioritized variants and confidently narrow down target genes in associated loci”.

While the authors did identify most likely causal variants at some of the loci through this method, they cannot claim to have confidently narrowed down target genes in associated loci through these methods.

We regret the lack of clarity of our initial sentence. By “public resources available in arteries” we were referring to available epigenetic datasets from ENCODE and Coronary arteries published by from T. Quertermous lab, in addition to eQTL datasets from the GTEx portal. All these datasets helped us narrow down target genes at FMD loci). We have now rephrased this statement to clarify this point (See L141-144).

The following claim is not valid. On what grounds/results is it made?:

We found that risk genes for FMD are specifically and consistently expressed in smooth muscle cells, fibroblasts, and arterial tissues and are involved in regulatory mechanisms related to actin cytoskeleton and intracellular calcium homeostasis, a mechanism central to vascular contraction.

For example, PHACTR1 and LRP1 are widely expressed (including brain, lung, whole blood, skin, adipose; see GTEx) not specifically in the mentioned tissues. Additionally, while you found that artery-specific ATAC-Seq peaks are overrepresented in the vicinity of genes involved in contractile fibres and muscle system processes in general, you did not establish through your analyses that FMD genes are involved in these processes.

We have now rephrased the problematic sentence to better reflect that the expression of FMD associated genes is not restricted to arteries, and that we did not provide new findings on their function. The involvement of *LRP1*, *ATP2B1* and *SLC24A3* (a.k.a NCKX3) in calcium homeostasis and artery contraction is well-established from previous work that we cite (see refs 34-37). The role of *PHACTR1* in actin cytoskeleton dynamics was previously reported as well (See ref 46).

Gene-based analysis:

If significant, the gene-based analysis only indicates the presence of signals physically mapping to genes, but the analysis does not provide direct evidence of these associated genes to be the biological target genes.

Thus, in RESULTS, listing up genes along with insignificant P-values resulting from the analysis has no meaning. Further, the results would not point to potential causal genes, even if they were significant! To avoid reader distraction, the reviewer suggests to shrink the paragraph in results (or even better to remove altogether) to one sentence saying something like the gene-based analyses using MAGMA did not identify additional association signals (refer to results in supplement).

We modified the text following the reviewer's suggestion (L171-172).

FMD associated variants are likely to regulate gene expressions of nearby genes in arterial tissues: In this results section, the lead variants at the PHACTR1, LRP1, and ATP2B1 loci are all correlated with the top cis-eQTL for these genes indicating that the FMD signals may be driven through an effect on expression of these genes. However, in the next results section "Transcriptome-wide association study in arteries" association between expression of PHACTR1 and LRP1 with the disease are confirmed, but the authors state that the association of ATP2B1 is not significant. What is the conclusion/interpretation then regarding ATP2B1?? Do the authors doubt that ATP2B1 is the causal gene behind the signal given these results?

To clarify our statement, we have changed the presentation of TWAS results to clearly draw the line between statistically significant associations (*LRP1*, *PHACTR1*, *SLC24A3*, presented in Table 2) and suggestive associations (*ATP2B1* and *ATF1*, in addition to other associations presented in Supplementary Table S5). The suggestive association of *ATP2B1* was mentioned to stress *ATP2B1* as the most likely causal gene at this locus. We have modified our text to match this purpose.

In table 2, the authors should present the lead SNP at the locus rather than presenting results for gene-based association. According to comment above, the signals resulting from gene-based analysis are not providing superior evidence (to the lead SNP) in support of the gene being the causal gene.

We have updated Table 2 and now only indicate the significant results from TWAS.

The following results are not very convincing (borderline significance), in fact they make little sense and only confuse the reviewer. Why would the variant only affect expression in patients but not in non-FMD patients? Consider removing this sentence. After all, expression of PHACTR1 associates with rs9349379 in non-FMD patients in GTex:

Interestingly, a query of transcriptome sequencing data obtained in primary dermal fibroblasts cell lines from 83 FMD patients and 70 matched controls found rs9349379 to correlate with PHACTR1 expression in the FMD group ($P=0.01$), but not the control group (Supplementary Figure S3).

Following this reviewer comment, we propose to mention only the association between the lead SNP and PHACTR1 expression in this locus that we found in fibroblasts from FMD patients, which is novel compared to GTex data, although we acknowledge that our sample is very modest ($N=83$). We have updated our text and Supplementary Figure S3 to restrict the description of data to FMD patients (L198-200).

FMD GWAS variants mapping to ATAC-Seq peaks in arterial primary cells and tissues

I find that these results are given too much space (and interpretation) given that they do not seem significant (P-values from 0.07 -0.01 for enrichment of lead variants that are not significant to begin with (restricting on $P<10^{-4}$ and correlated variants with $r^2>0.7$). I strongly recommend to shrink this section considerably (perhaps move to Supplement), or even better, to remove altogether.

We followed the reviewer's suggestion and moved this part to supplementary results.

FMD potential target genes are expressed in VSMCs and fibroblasts

I think this section should be shortened considerably – the message being that the PHACTR1, LRP1, and other candidate genes are expressed in tissues that are relevant for FMD.

As suggested, we reduced and simplified the text to provide a more straight-to-the-point description of these results (L264-273). We grouped Supplementary Figures S6 and S7 (now S7) and deleted Supplementary Figure S8 to match with the text.

Further, the reviewer finds these results somewhat contradicting the results where the authors claim to have found enrichment (although not highly significant) of lead variants in open chromatin peaks of artery tissues, but NOT in SMC and fibroblasts.

We agree that these observations may seem partly contradictory; this is in fact one of the main reasons why we wanted to indicate the expression of potential FMD-associated genes based on single-cell experiments. There is little doubt that observations made in artery tissue, such as eQTL associations or DNA accessibility, are largely due to vascular smooth muscle cells and fibroblasts, which together represent 50-80% of the total cell content in human arteries (see for example Hu et al., *ATVB* 2021, DOI: 10.1161/ATVBAHA.120.315373, Zhang et al., *MedRxiv* 2021, DOI: 10.1101/2021.02.17.431699). However, we observe strong differences between artery tissue and cultured smooth muscle cells, especially at the regulatory level. We could expect that these differences would reflect the influence of other cell types in artery tissue. Nonetheless, the comparison of open chromatin regions in artery tissue and smooth muscle cells rather suggests that most of these differences are due to the fact cultured SMCs lose part of their contractile function. For these reasons, we sought to confirm that FMD associated genes were reported to be expressed in vascular smooth muscle cells and fibroblasts when directly extracted from artery tissue without ongoing cell culture.

I am further sceptic about whether the potential association of candidate gene expression with sex can be considered significant, and I would certainly not suggest that PHACTR1 and ATF1 had sex-specific expression. These results can be removed without harming the overall story.

Following the reviewer comment, we removed the suggestive results (see L269-270). We only keep the results regarding *SLC24A3* that is more robust and consistent with previous observations suggesting that this gene is highly regulated by estrous cycle in rodents (see Yang et al., *Mol Biol Reprod* 2010, DOI: 10.1002/mrd.21245), and differently expressed in mouse kidney (Lee et al., *Am J Physiol Renal Physiol* 2009, DOI: 10.1152/ajprenal.90615.2008). This information was added in the discussion (L447-448).

Discussion:

Use a more descriptive wording for the following: “ii) through integration of annotation datasets generated inhouse”, i.e. in what house was it generated?

Please revise/reconsider the following claim (see comment above):

iii) FMD risk genes showed consistent and specific expressions in VSMCs, fibroblasts and arterial tissue and were involved in regulatory mechanisms related to actin cytoskeleton and intracellular calcium homeostasis, a mechanism central to vascular contraction.

We modified these sentences as suggested by the reviewer (L342-347).

Reviewers' Comments:

Reviewer #2:

None